# BANDIT LEARNING IN MATCHING MARKETS ROBUST TO ADVERSARIAL CORRUPTIONS

**Zheshun Wu**[1,2]    **Jinhang Zuo**[3]    **Zenglin Xu**[4,5]    **Fang Kong**[1]*
[1]Southern University of Science and Technology    [2]Harbin Institute of Technology, Shenzhen
[3]City University of Hong Kong    [4]Fudan University    [5]Shanghai Academy of AI for Science
{wuzhsh23,jinhangzuo,zenglin}@gmail.com   kongf@sustech.edu.cn

## ABSTRACT

This paper investigates the problem of bandit learning in two-sided decentralized matching markets with adversarial corruptions. In matching markets, players on one side aim to learn their unknown preferences over arms on the other side through iterative online learning, with the goal of identifying the optimal stable match. However, in real-world applications, stochastic rewards observed by players may be corrupted by malicious adversaries, potentially misleading the learning process and causing convergence to a sub-optimal match. We study this problem under two settings: one where the corruption level $C$ (defined as the sum of the largest adversarial alterations to the feedback across rounds) is known, and another where it is unknown. For the known corruption setting, we develop a robust variant of the classical Explore-Then-Gale-Shapley (ETGS) algorithm by incorporating widened confidence intervals. For the unknown corruption case, we propose a Multi-layer ETGS race method that adaptively mitigates adversarial effects without prior corruption knowledge. We provide theoretical guarantees for both algorithms by establishing upper bounds on their optimal stable regret, and further derive the lower bound to demonstrate their optimality.

## 1 INTRODUCTION

Two-sided matching markets have garnered significant research attention due to their central role in marketplace applications (Gale & Shapley, 1962), from crowdsourcing markets (matching customers with freelancers) (Sun et al., 2023) to ridesharing platforms (pairing passengers with riders) (Dickerson et al., 2021). In these markets, each participant maintains a preference ordering over the other side (Liu et al., 2020), derived from latent utilities such as a freelancer's service quality or a rider's reliability. One of the core evaluation metrics on matching markets is stability (Roth & Sotomayor, 1992), which illustrates equilibrium states where no participant pair can mutually benefit from deviating from their current match.

In real-world applications, for example, labor markets with employers and workers, employers are not aware of the real working qualities of workers before employment. Hence, their preferences over workers are unclear (Liu et al., 2020; Kong & Li, 2023). A key challenge arises on how to attain an optimal stable matching among competing participants, only through learning from the iterative matchings with the other side (Liu et al., 2021). As a well known learning framework under uncertainty, multi-armed bandit (MAB) has been widely used in many sequential decision-making applications (Auer et al., 2002; Slivkins, 2020). In recent years, the MAB framework has been studied in many studies of matching markets (Liu et al., 2020; 2021; Basu et al., 2021; Kong et al., 2024; 2025). These works regard players and arms in MAB as two sides of the market participants. Each player has unknown preferences over arms corresponding to unknown reward distributions in MAB. Hence, players aim to learn the distribution iteratively via collecting empirical observations to minimize the player-optimal regret defined as comparing practical matching with the players' most-preferred stable matching. We notice that the common bandit learning model for matching markets significantly depends on stochastic rewards. In words, most of the current works assume that

---

*Corresponding author.

the rewards generated by successful matchings between players and arms are drawn from unknown but fixed distributions.

More specifically, existing bandit algorithms in matching markets commonly assume that the feedback players receive from arms follows the true preference model (Liu et al., 2020; Kong & Li, 2023; Zhang & Fang, 2024). However, this assumption is often difficult to satisfy in real-world scenarios. External noise or deliberate manipulation may compromise the authenticity of the feedback. For example, in the server resource allocation problem (Hussain et al., 2013), certain computing nodes may exhibit performance far exceeding their actual capabilities during the testing phase to induce the system to allocate more resources (Pamarthi & Narmadha, 2022). Click fraud in advertising scenarios is another typical example of feedback contamination (Zhang & Guan, 2008; Oentaryo et al., 2014), where competitors or malicious third parties may use automated scripts or click farms to generate a large number of clicks, creating false click-through rates. These interferences can severely distort feedback signals, preventing players from accurately learning the true preferences. Yet, existing algorithms lack defense mechanisms against contaminated feedback, rendering them unable to converge to the true stable matching once the feedback is polluted.

We observe that these challenges can be naturally modeled as a stochastic multi-armed bandit (MAB) problem with adversarial corruptions (Lykouris et al., 2018; Gupta et al., 2019), where each arm pull produces a stochastic reward that may be perturbed by an adversary before being revealed to the player. We further show that standard bandit algorithms for matching markets with stochastic feedback, such as the explore-then-Gale-Shapley (ETGS) algorithm (Kong & Li, 2023), are highly vulnerable: even limited adversarial corruption can mislead players into consistently matching with suboptimal arms, resulting in linear regret. These insights motivate us to design new algorithms for matching markets that remain robust under adversarial corruptions.

In this paper, we take the first step to study a bandit learning problem of decentralized matching markets with adversarial corruptions. We first provide a robust variant of ETGS with widened confidence intervals for tackling the known corruption setting as a warm-up. For the unknown corruption setting, we devise a Multi-layer ETGS race method that can handle any level of corruption, and its performance degrades gracefully as more corruption is added. We highlight that this method can both tolerate corrupted feedback and utilize the stochastic component of feedback to enhance bandit learning. However, incorporating existing randomized algorithms (Lykouris et al., 2018) into ETGS leads to frequent matching conflicts, resulting in inefficient feedback collection. To address this challenge, we introduce a sub-phase level synchronization mechanism to avoid conflicts, and develop a novel martingale concentration inequality to design principled confidence intervals. We also uncover an intrinsic trade-off between communication cost and learning efficiency in the unknown corruption setting, and further propose a joint optimization strategy to identify the optimal synchronization interval for balancing this trade-off. Eventually, we establish the regret lower bound of this problem to demonstrate the tightness of our algorithms' regret upper bounds.

Our contributions can be summarized as follows:

- Our work investigates a new bandit learning problem of decentralized matching markets with adversarial corruptions, capturing more practical adversarial scenarios.

- We observe that directly extending existing randomized algorithms for matching markets with unknown corruption leads to frequent conflicts in the vanilla ETGS algorithm. To overcome this challenge, we develop a novel Multi-layer ETGS race method with a synchronization mechanism that coordinates effective exploration among the players.

- We derive a novel martingale concentration inequality tailored for the synchronization mechanism to bound the total corruption suffered by Multi-layer ETGS race with high probability. This inequality allows us to set a principled confidence radius. Finally, we reveal a fundamental trade-off between communication overhead and learning efficiency in the unknown corruption setting, and further identify the optimal synchronization interval that balances this trade-off.

- We prove player-optimal stable regret upper bounds for proposed algorithms, and further provide the lower bound to show their optimality.

Due to the space limit, more related works can be found in Appendix A.

## 2 PRELIMINARIES

In this section, we provide the problem formulation of bandit learning in matching markets robust to adversarial corruptions.

**Two-sided matching markets.** We consider $N$ players and $K$ arms, and assume $N \leq K$. Define $\mathcal{N} = \{p_1, p_2, \ldots, p_N\}$ be the player set and $\mathcal{K} = \{a_1, a_2, \ldots, a_K\}$ be the arm set. We describe the preference rank of player $p_i$ over arm $a_j$ by a real value $\mu_{i,j} \in (0, 1]$. We can notice that the greater value of $\mu_{i,j}$ demonstrates more preference on arm $a_j$. Without loss of generality, we consider that all preference lists are heterogeneous, i.e., $\mu_{i,j} \neq \mu_{i,j'}$ for distinct arms $a_j \neq a_{j'}$, keeping consistent to previous works (Sankararaman et al., 2021; Kong & Li, 2023). Besides, each arm is equipped with a preference ranking over players. Denote $(\pi_{j,i})_{i \in [N]}$ as the distinct preference values of arm $a_j$ over players. Then $\pi_{j,i} > \pi_{j,i'}$ implies $a_j$ prefers $p_i$ to $p_{i'}$. Motivated by real applications such as online labor market Upwork with employers and workers, the preferences of players are usually uncertain and can be learnt through iterative matching processes. While arms usually know their preferences based on some known utilities such as the payment of employers. In matching markets, stability is a key concept (Abdulkadiroğlu & Sönmez, 2013). Formally, the matching $\bar{A}(t)$ is stable if no player and arm has incentive to abandon their current partner, i.e., there exists no player-arm pair $(p_i, a_j)$ such that $\mu_{i,j} > \mu_{i,\bar{A}_i(t)}$ and $\pi_{j,i} > \pi_{j,\bar{A}_j^{-1}(t)}$, where we simply define $\pi_{j,\emptyset} = -\infty$ and $\mu_{i,\emptyset} = -\infty$ for each $j \in [K], i \in [N]$. Notice that there may be more than one stable matching in the market.

**Bandit learning in matching markets.** In each round $t \in [T]$, each player $p_i$ proposes to an arm $A_i(t) \in \mathcal{K}$. Correspondingly, each arm $a_j$ receives requests from players in $A_j^{-1}(t) := \{p_i : A_i(t) = a_j\}$. Analogous to the labor market where a worker can only work for one task, arms would only accept one request from the player that it prefers most. If a player $p_i$ is successfully accepted by the proposed arm $\bar{A}_i(t)$, it will receive a random reward $X_i(t) = r_{i,\bar{A}_i(t)}^S(t) \sim \mathcal{F}_{i,\bar{A}_i(t)}$ corresponding to the expectation characterizing its matching experience in this round, which we assume is 1-subgaussian with expectation $\mu_{i,\bar{A}_i(t)}$. And if $p_i$ is rejected, it only receives $X_i(t) = 0$. For convenience, denote $A(t) = \{(i, A_i(t)) : i \in [N]\}$ as the selections of all players and $\bar{A}(t) = \{(i, \bar{A}_i(t)) : i \in [N]\}$ as the final matching at round $t$.

**Matching markets with corrupted feedback.** In this paper, we consider that there exists an adaptive adversary who can corrupt certain stochastic rewards acquired by players based on historical information. In the following, we take the player $p_i$ as an example to illustrate the interaction process between the player and adversary. The interaction protocol is formally provided as follows,

1. For player $p_i$, a stochastic reward $r_{i,j}^S(t)$ is drawn for each arm $j \in [K]$ according to the reward distribution $\mathcal{F}_{i,j}$ with mean $\mu_{i,j}$.

2. For any arm $a_j, j \in [K]$, the adversary observes the realizations of $r_{i,j}^S(t)$, along with rewards and matches of player $p_i$ in previous rounds. Then the adversary returns a corrupted reward $r_{i,j}(t) \in [0, 1]$.

3. If $p_i$ is matched successfully with its proposed arm $A_i(t)$, $p_i$ will observe the corresponding corrupted feedback $X_i(t) = r_{i,A_i(t)}(t)$.

Similar to Lykouris et al. (2018), we define $\max_{j \in [K]} |r_{i,j}(t) - r_{i,j}^S(t)|$ as the amount of corruption injected in round $t$ for player $p_i$. The level of total corruption incurred by $p_i$ is defined as

$$\sum_{t \in [T]} \max_{j \in [K]} \left| r_{i,j}(t) - r_{i,j}^S(t) \right| \leq C. \tag{1}$$

We emphasize that the adversary is allowed to be adaptive, i.e., the corruptions on round $t$ can be chosen as a function of the past matches and stochastic rewards of player $p_i$.

In this paper, we consider one standard metric in stochastic MAB termed *pseudo regret*. Furthermore, the player-optimal stable pseudo regret is considered for the matching market setting. In specific, let $M := \{m : m \text{ is a stable matching}\}$ be the set of all stable matchings and $m^* = \{(i, m_i^*) : i \in [N]\} \in M$ be the players' most preferred one. That is to say, $\mu_{i,m_i^*} \geq \mu_{i,m_i}$ for any $m \in M, i \in [N]$. Our objective is to learn the player-optimal stable matching $m^*$ and minimize

the player-optimal stable pseudo regret for each $p_i \in \mathcal{N}$, which is defined as the cumulative reward difference between being matched with $m_i^*$ and that $p_i$ receives over $T$ rounds:

$$Reg_i(T) = \sum_{t=1}^{T} \mu_{i,m_i^*} - \mathbb{E}\left[\sum_{t=1}^{T} X_i(t)\right], \tag{2}$$

where the expectation is taken over the randomness in the received rewards, the players' algorithmic decisions, and the adversary's strategy.

For convenience, we also introduce several notations to measure the hardness of the bandit learning problem in matching markets, which will be used in the later analysis.

**Definition 2.1.** *For each player $p_i$ and arm $a_j \neq a_{j'}$, let $\Delta_{i,j,j'} = |\mu_{i,j} - \mu_{i,j'}|$ be the preference gap of $p_i$ between $a_j$ and $a'_j$. Let $\rho_i$ be player $p_i$'s preference ranking and $\rho_{i,k}$ be the $k$-th preferred arm in $p_i$'s ranking for $k \in [K]$. Define $\Delta = \min_{i \in [N]; k,k' \in [N+1]; k \neq k'} \Delta_{i,\rho_{i,k},\rho_{i,k'}}$ as the minimum preference gap among all players and their first $N + 1$-ranked arms, which is non-negative since all preferences are distinct. Further, for each player $p_i$, let $\Delta_{i,\max} = \mu_{i,m_i^*}$ be the maximum player-optimal stable regret that may be suffered by $p_i$ in all rounds.*

## 3 ALGORITHM

In this section, we first analyze the inherent vulnerability of the ETGS algorithm (Kong & Li, 2023) under adversarial corruptions. We then demonstrate a fundamental limitation of ETGS: introducing randomness into ETGS for mitigating corruption will inevitably lead to frequent matching conflicts. To address these challenges, we propose the main algorithm in this paper termed Multi-layer ETGS race method, which maintains multiple ETGS instances at different learning rates to achieve robustness against any level of unknown corruptions. To eliminate matching conflicts arising from the proposed algorithm's randomness, we develop a sub-phase level synchronization mechanism that coordinates all players to stay in a same ETGS instance by an elected leader. Besides, we provide a robust variant of ETGS with widened confidence intervals, which is used for tackling the known corruption setting, as a warm-up for the Multi-layer ETGS race method.

**Vulnerability of ETGS.** The core idea of ETGS is to collect sufficient observations in a Round-Robin manner to accurately estimate preference rankings and then attain the optimal stable matching via the offline Gale-Shapley algorithm. However, ETGS exhibits significant vulnerability under adversarial corruptions. Specifically, an adaptive adversary can observe the information of all the past matches $\bar{A}_i(t)$ and the corresponding stochastic feedback $r_{i,A_i(t)}^S$ of $p_i$ to execute corruption injection in the current round. Consequently, since the arm proposing follows a deterministic Round-Robin pattern, the adversary can systematically corrupt the optimal arm for player $p_i$. This forces $p_i$ to persistently match with a suboptimal arm, misleading its learning process. Such manipulation requires only $\mathcal{O}(\log T/\Delta^2)$ rounds to take effect, ultimately inflicting $\mathcal{O}(T)$ cumulative regret on $p_i$.

**Frequent conflicts in original ETGS caused by algorithmic randomness.** Introducing randomness is a well-established technique for combating adversarial corruptions in bandit algorithms (Lykouris et al., 2018; Gupta et al., 2019). Randomized algorithms commonly provide inherent tolerance compared to deterministic approaches like UCB or arm elimination (Lattimore & Szepesvári, 2020). However, in decentralized matching markets, when players independently employ randomized strategies, the probability of matching conflicts increases substantially, leading to inefficient exploration and degraded learning performance eventually.

Based on the above observations, we develop robust bandit learning algorithms for decentralized matching markets under adversarial corruptions. Our proposed algorithm proceeds through three sequential phases: (1) each player first identifies a unique self-identifier; (2) through strategic exploration, players estimate their preference rankings over the top $N$ arms; and (3) leveraging these estimates, they identify and permanently match with their optimal stable arm in all subsequent rounds. The key distinction between our algorithm and ETGS emerges in the second phase. Specifically, we introduce a Multi-layer ETGS race method to achieve robustness under agnostic corruption. In the following sections, we detail these bandit learning innovations for Phase 2, while deferring the workflows of Phase 1 and Phase 3 in Appendix B. Here we provide some insights about Phase 1 and Phase 3. Phase 1 is a $N$-round iterative process that assigns a distinct index to each player based on

their acceptance by a single preference $a_1$. Phase 3 can be regarded as a decentralized Gale-Shapley algorithm (Gale & Shapley, 1962), where the objective is for players to find their respective arm in the optimal stable matching based on estimated ranking.

Prior to formally introducing our main algorithm, termed the Multi-layer ETGS race method, we first present a robust ETGS variant designed for known corruption settings as a warm-up. This preliminary algorithm enlarges confidence intervals in proportion to the corruption level $C$, establishing the foundation for our subsequent developments.

## 3.1 ALGORITHM WITH KNOWN CORRUPTION LEVEL

As introduced earlier, all players enter in the second phase after identifying their distinct indices during the first phase. Similar to ETGS, we consider that the second phase is further divided into several sub-phases $k = 1, 2, \ldots$ with the corresponding lengths denoted by $d_1, d_2, \ldots$ respectively. For the known corruption setting, we adopt the same sub-phase structure as ETGS, where each sub-phase $k$ consists of $2^k + 1$ rounds. Specifically, sub-phase $k$ begins with an exploration stage of length $2^k$ rounds and concludes with a single monitoring round. This monitoring round is used for checking whether the preferences of players are estimated well, and is introduced in Appendix B in detail. During this exploration phase, players aim to collect sufficient feedback on arms and detect whether the preference ranks of all players have been estimated well in the monitoring round.

During each sub-phase of the second phase, players propose to arms in a round-robin fashion. Leveraging the unique indices obtained in Phase 1, distinct players are guaranteed to pick different arms, ensuring conflict-free matching with their proposed arms. After $p_i$ obtains the feedback on the matched arm $A_i(t)$, it would update the estimated preference value $\hat{\mu}_{i,A_i(t)}$ and the observed time $T_{i,A_i(t)}$ for this selected arm $A_i(t)$ as

$$\hat{\mu}_{i,A_i(t)} = \left(\hat{\mu}_{i,A_i(t)} \cdot T_{i,A_i(t)} + X_{i,A_i(t)}(t)\right) / \left(T_{i,A_i(t)} + 1\right), \ T_{i,A_i(t)} = T_{i,A_i(t)} + 1 \,.$$

At the end of this second phase, players would construct a confidence interval for each estimated preference value based on its previously collected feedback. Given the prior information of corruption level $C$, one intuitive idea is to directly enlarge the confidence intervals to handle the worst-case corruption. Recall that the corrupted rewards $r_{i,A_i(t)}(t)$ can be decomposed into two terms $r^{\mathcal{S}}_{i,A_i(t)}(t) + c_{i,A_i(t)}(t)$, where the second term $c_{i,A_i(t)}(t)$ denotes the corruption injected by the adversary in this round. Thus, if the total corruption introduced by the adversary is at most $C$, the confidence interval of $p_i$ for the preference value over $a_j$ can be established with the upper bound UCB and lower bound LCB defined as

$$\mathrm{UCB}_{i,j} = \hat{\mu}_{i,j} + \sqrt{\frac{6 \log T}{T_{i,j}}} + \frac{C}{T_{i,j}}, \quad \mathrm{LCB}_{i,j} = \hat{\mu}_{i,j} - \sqrt{\frac{6 \log T}{T_{i,j}}} - \frac{C}{T_{i,j}}, \quad (3)$$

where we simply let $\mathrm{UCB}_{i,j}$ be $\infty$ and $\mathrm{LCB}_{i,j}$ be $-\infty$ when $T_{i,j} = 0$. When the confidence sets for two arms $a_j, a_{j'}$ are disjoint, i.e., $\mathrm{LCB}_{i,j} > \mathrm{UCB}_{i,j'}$ or $\mathrm{LCB}_{i,j'} > \mathrm{UCB}_{i,j}$, $p_i$ can determine its preference over these arms.

Then we present the upper bound for the player-optimal stable pseudo regret of each player by our algorithm. The corresponding proof is provided in Appendix C.

**Theorem 3.1.** *Following the Algorithm 1, the player-optimal stable pseudo regret of each player $p_i \in \mathcal{N}$ satisfies*

$$Reg_i(T) \le \left(N + 8K\left(\frac{48 \log T}{\Delta^2} + \frac{C}{\Delta}\right) + \log\left(8K\left(\frac{48 \log T}{\Delta^2} + \frac{C}{\Delta}\right)\right) + N^2 + 2NK\right) \cdot \Delta_{i,\max}$$
$$= \mathcal{O}\left(K \log T / \Delta^2 + KC/\Delta\right).$$
$$(4)$$

**Remark 3.2.** *The regret upper bound of our proposed algorithm for known corruption setting is similar to that of Kong & Li (2023). The first term in Eq. (4) is the upper bound for regret incurred in phase 1, the second term is the regret upper bound for the total exploration rounds and the third term is the upper bound for the total monitoring rounds in phase 2, the fourth term is the regret upper bound for phase 3 and the last constant term corresponds to the bad concentration events.*

---

**Algorithm 1** Robust ETGS with Widened Confidence Intervals (from view of player $p_i$)

---

1: Input: total corruption $C$, player set $\mathcal{N}$, arm set $\mathcal{K}$, horizon $T$
2: Initialize: $\hat{\mu}_{i,j} = 0, T_{i,j} = 0, \forall j \in [K]$
3: Phase 1: the index estimation phase in ETGS
4: //Phase 2, learn the preferences
5: **for** $k = 1, 2, \ldots$ **do**
6:      $\mathrm{F}_k =$ False //whether the preference has been estimated well
7:      **for** $t = N + \sum_{k'=1}^{k-1}(2^{k'} + 1) + 1, \ldots, N + \sum_{k'=1}^{k-1}(2^{k'} + 1) + 2^k$ **do**
8:          $A_i(t) = a_{(\text{Index}+t-1)\%K+1}$
9:          Observe $X_{i,A_i(t)}(t)$ and update $\hat{\mu}_{i,A_i(t)}, T_{i,A_i(t)}$ if $\bar{A}_i(t) = A_i(t)$
10:      $t_k \leftarrow N + \sum_{k'=1}^{k-1}(2^{k'} + 1) + 2^k$
11:      Compute $\text{UCB}_{i,j}$ and $\text{LCB}_{i,j}$ for each $j \in [K]$ defined in Eq. (3)
12:      $\sigma_i, O_k \leftarrow$ Monitoring($\{\text{UCB}_{i,j}, \text{LCB}_{i,j}\}_{j \in [K]}, t_k$)
13:      **if** $|O_k| == N$ **then**
14:          Enter in the next phase with $\sigma_i$
15: Phase 3: the phase of identifying optimal stable match via decentralized GS algorithm in ETGS.
     To find the optimal stable arm with $\sigma_i = (\sigma_{i,1}, \sigma_{i,2}, \ldots, \sigma_{i,K})$

---

## 3.2 Algorithm with Unknown Corruption Level

In the previous subsection, we aim to make the algorithm robust to corruption by directly enlarging confidence intervals. However, this method is inapplicable to unknown corruption settings, as confidence intervals cannot be properly calibrated without knowledge of the corruption level $C$. In this section, we devise a Multi-layer ETGS race method for achieving robustness under the unknown corruption setting. We highlight that the core innovation of this method includes: (1) handling all possible amounts of corruption via maintaining multiple ETGS instances with different levels of robustness; (2) establishing a sub-phase level synchronization mechanism for avoiding frequent matching conflicts caused by algorithmic randomness, and further identifying the optimal synchronization interval to balance the trade-off between communication cost and learning efficiency.

Similar to Lykouris et al. (2018), we introduce $\log T$ instances of ETGS to address the agnostic corruption. Specifically, we assign each instance a distinct sampling probability, and probabilistically select ETGS instances during the bandit learning. Intuitively, instances with lower sampling probabilities are statistically exposed to fewer corruptions, thereby achieving stronger robustness. However, a critical challenge arises when players independently sample ETGS instances: frequent matching conflicts occur because players may select different algorithm instances in a given round, rendering the Round-Robin mechanism in the original ETGS ineffective. To resolve this, we introduce a synchronization mechanism wherein a unique leader is elected before bandit learning. This leader exclusively performs random sampling of ETGS instances at the end of each sub-phase and broadcasts the selected instance index to other players through arm pulls, ensuring all players operate on the same ETGS instance. This mechanism effectively eliminates potential matching conflicts. The details of leader selection and the synchronization mechanism are deferred to Appendix B. Below, we outline the core idea of our mechanism. The leader selection can be implemented by designating player 1, who first obtains an index, as the leader at the end of Phase 1. To achieve synchronization across players, the leader can propose a specific arm in a predetermined future round at the end of each sub-phase, which is used to convey its selected layer index to the other players.

**Trade-off between communication overhead and learning efficiency.** For the unknown corruption setting, we also divide the exploration phase into several sub-phases. The difference is that the length of each sub-phase is set to a constant $d$. At the end of each sub-phase, the leader selects the $\ell$-th ETGS instance with the probability proportional to $2^{-\ell}$, and broadcasts the index of the selected layer to other players by pulling arms. Then in the next sub-phase, players would propose arms based on the corresponding Round-Robin pattern of the selected ETGS instance, thereby avoiding conflicts. Intuitively, when the constant $d$ is set to 1, the proposed sampling strategy recovers per-round sampling used in Lykouris et al. (2018). It means the leader should communicate with other players in each round to achieve synchronization, resulting in a severe communication cost. While increasing $d$ reduces communication overhead, it may amplify the corruptions experienced

by the selected ETGS instance within each sub-phase (as formally analyzed in Lemma 3.4). **This reveals a fundamental trade-off between the communication overhead and learning efficiency in setting hyper-parameter $d$.** Below, we minimize the regret bound w.r.t. $d$ to select the optimal hyper-parameter. Notice that observations collected in each sub-phase are solely used to update the statistics of the corresponding ETGS instance. Below we show that if the corruption level is at most $C$, then instances with $\ell \geq \log C$ will observe at most $\mathcal{O}(d \log T)$ corruption with high probability.

**Multi-layer ETGS race.** The proposed algorithm is called *race* since we regard it as multiple ETGS instances racing to estimate the accurate preference ranks. If the $\ell$-th layer ETGS has estimated preferences well for all players based on its own statistics, players will run the offline GS algorithm to identify the optimal stable match for this layer. In the remaining rounds, if one layer, whose optimal stable match has been identified, is selected in a sub-phase, players would propose to arms following this optimal stable match in this sub-phase. Besides, if the $\ell$-th layer has finished identifying its optimal stable match $\sigma^\ell$, then optimal stable matches $\sigma^{\ell'}$ of all layers $\ell' \leq \ell$ would be modified to be the same as $\sigma^\ell$. This is because the ETGS instance with a lower sampling probability is slower but more precise. Intuitively, instances with lower sampling probabilities are affected by less corruption. In the following, we provide a rigorous analysis of the fact that instances behave as without any corruption if they are selected with the sampling probability lower than $1/C$.

**Technical challenge.** As previously introduced, our synchronization mechanism resolves matching conflicts by delegating layer sampling to a unique leader at the end of each sub-phase. However, this algorithm design renders the martingale concentration inequality from Lykouris et al. (2018), which is developed for per-round bandit instance sampling, inapplicable to our algorithm. To address this challenge, we derive a sub-phase level martingale concentration inequality that establishes the high-probability bound on the total corruption observed by the ETGS instance with sampling probability $1/C$. Based on this inequality, we can design reasonable confidence intervals below. Below we provide a lemma to show that the amount of corruption that actually affects the layer with sampling probability $1/C$ is at most $\mathcal{O}(d \log T)$.

**Lemma 3.3.** *In Algorithm 2, the ETGS instance with sampling probability less than $1/C$ experiences, w.p. at least $1 - 1/T$, cumulative corruption bounded by $d \log(T) + 2$ during the exploration phase.*

We provide the detailed proof of this lemma in Appendix D.

---

**Algorithm 2** Multi-layer ETGS race (from view of player $p_i$)

---

1: Input: player set $\mathcal{N}$, arm set $\mathcal{K}$, horizon $T$, the fixed length $d$ for any sub-phases $k$ in Phase 2
2: Initialize: $\hat{\mu}_{i,j}^\ell = 0, T_{i,j}^\ell = 0, \forall j \in [K], \forall \ell \in [\log T]$
3: Phase 1: the index estimation phase in ETGS
4: //Phase 2, learn the preferences
5: **for** $k = 1, 2, \ldots$ **do**
6:     $\mathrm{F}_k^\ell = $ False //whether the preference of instance $\ell$ has been estimated well
7:     **for** $t = N + \sum_{k'=1}^{k-1}(d + 1 + c_{k'}) + 1, \ldots, N + \sum_{k'=1}^{k-1}(d + 1 + c_{k'}) + d$ **do**
8:         Select $A_i(t)$ based on the corresponding Round-Robin pattern of layer $\ell$
9:         Observe $X_{i,A_i(t)}(t)$ and update $\hat{\mu}_{i,A_i(t)}^\ell, T_{i,A_i(t)}^\ell$ if $\bar{A}_i(t) = A_i(t)$
10:    $t_k \leftarrow N + \sum_{k'=1}^{k-1}(d + 1 + c_{k'}) + d$
11:    Compute $\mathrm{UCB}_{i,j}^\ell$ and $\mathrm{LCB}_{i,j}^\ell$ for each $j \in [K]$ defined in Eq. (5).
12:    $\sigma_i^\ell, O_k^\ell \leftarrow$ Monitoring($\{\mathrm{UCB}_{i,j}^\ell, \mathrm{LCB}_{i,j}^\ell\}_{j \in [K]}, t_k$)
13:    **if** $|O_k^\ell| == N$ **then**
14:        Find the optimal stable arm with $\sigma_i^\ell = (\sigma_{i,1}^\ell, \sigma_{i,2}^\ell, \ldots, \sigma_{i,K}^\ell)$ via the decentralized offline GS algorithm, and set the optimal stable matches $\sigma^{\ell'}$ of all the layers $\ell' \leq \ell$ the same as $\sigma^\ell$
15:    **if** $p_i$ is the leader **then**
16:        Sample layer $\ell \in [\log T]$ with probability $2^{-\ell}$. With remaining prob, sample $\ell = 1$.
17:        Communicate $\ell$ to other players via pulling arms in $c_k = \lfloor \ell \rfloor$ rounds

---

We then formally introduce the proposed Multi-layer ETGS race algorithm for the unknown corruption setting. This algorithm maintains $\ell = 1 \ldots \log T$ different instances of ETGS. Each instance $\ell \in [\log T]$ keeps its own empirical mean $\hat{\mu}_{i,j}^\ell$ corresponding to the average empirical reward of the match between $p_i$ and $a_j$, and also keeps track of how many times $a_j$ was matched with $p_i$ in this

instance $T_{i,j}^\ell$. At the end of each sub-phase, the elected leader samples $\ell \in [\log T]$ with probability $2^{-\ell}$ and broadcasts $\ell$ to other players by pulling arms. Based on Lemma 3.3, we can define the same width of the confidence interval for $p_i$ and $a_j$ in the $\ell$-th layer as $\sqrt{\frac{6 \log T}{T_{i,j}^\ell}} + \frac{d \log T + 2}{T_{i,j}^\ell}$. By properly selecting $d$ we can ensure that $d \log T + 2 \le 2d \log T$, and the corresponding UCB and LCB for each layer $\ell$ can be thus defined as

$$\text{UCB}_{i,j}^\ell = \hat{\mu}_{i,j}^\ell + \sqrt{\frac{6 \log T}{T_{i,j}^\ell}} + \frac{2d \log T}{T_{i,j}^\ell}, \quad \text{LCB}_{i,j}^\ell = \hat{\mu}_{i,j}^\ell - \sqrt{\frac{6 \log T}{T_{i,j}^\ell}} - \frac{2d \log T}{T_{i,j}^\ell}. \quad (5)$$

Then we provide the regret guarantee for Algorithm 2 and defer its proof to Appendix E.

**Theorem 3.4.** *When configured with confidence intervals of width $\sqrt{\frac{6 \log T}{T_{i,j}^\ell}} + \frac{2d \log T}{T_{i,j}^\ell}$, the Multi-layer ETGS race algorithm achieves the player-optimal stable pseudo regret of each player $p_i \in \mathcal{N}$*

$$Reg_i(T) \le \underbrace{(N + N^2 \log T + 2NK \log T + N \log T)\Delta_{i,\max}}_{\textbf{Term (a)}}$$

$$+ \underbrace{\left( 16K \log T \left( \frac{24 \log T}{\Delta^2} + \frac{d \log T}{\Delta} \right) + 16KC \left( \frac{24 \log T}{\Delta^2} + \frac{d \log T}{\Delta} \right) \right) \cdot \Delta_{i,\max}}_{\textbf{Term (b)}}$$

$$+ \underbrace{16K \log T \left( \frac{24 \log^2 T}{d\Delta^2} + \frac{\log^2 T}{\Delta} + \frac{24C \log T}{d\Delta^2} + \frac{C \log T}{\Delta} \right) \cdot \Delta_{i,\max}}_{\textbf{Term (c)}}$$

$$\le \mathcal{O} \left( \frac{Kd \log T (\log T + C)}{\Delta} + \frac{K \log^2 T (\log T + C)}{d\Delta^2} + \frac{K \log^2 T (\log T + C)}{\Delta} \right). \quad (6)$$

**Remark 3.5.** *We first explain each term in this upper bound. Term (a) captures the regret incurred in phases 1 and 3, as well as from bad concentration events. Term (b) reflects the regret due to exploration within each sub-phase. Term (c) accounts for the regret arising from the total communication cost between sub-phases. As previously introduced, we select the optimal hyperparameter $d$ to minimize this regret upper bound. In specific, we set $d$ to be $\mathcal{O}(\sqrt{\log T})$, and the upper bound becomes $\mathcal{O}(K \log^{1.5} T (\log T + C)/\Delta^2 + K \log^2 T (\log T + C)/\Delta)$. When $C = 0$, we know that all ETGS instances finish their respective exploration phase with true optimal stable matches. For this scenario, the regret is at most $\mathcal{O}(K \log^{2.5} T/\Delta^2 + K \log^3 T/\Delta)$. The additional multiplicative $\log^2 T$ term is from the required rounds of finishing the exploration of $\log T$ ETGS instances and $\mathcal{O}(\log T)$ communication cost to acheive synchronization. When $N = 1$, there is no need to use the synchronization mechanism for avoiding matching conflicts. Thus we know that the regret is at most $\mathcal{O}(K \log T (\log T + C)/\Delta)$ at this time.*

*Proof sketch.* For the layer $r \in [\log T]$ whose sampling probability satisfies $2^{-r} \le 1/C$, the corruption it experiences is at most $\mathcal{O}(d \log T)$ with high probability. We can thus regard these layers as robust with purely stochastic feedback. The remaining problem is to bound the regret contributed by layers that are not robust to the corruption. We know that there exists some layer $\ell^*$ satisfying $\ell^* = \arg\min_\ell[2^\ell > C]$. According to our algorithm, when its stable match $\sigma^{\ell^*}$ is identified, the stable matches $\sigma^{\ell'}$ of layers $\ell' < \ell^*$ would be replaced with $\sigma^{\ell^*}$ and thus there is no regret for these faster layers in the remaining rounds. In expectation the sub-optimal arm is played as most $C$ times more in faster layers $\ell'$ compared with that of the layer $\ell^*$, and we can bound the regret contributed by layers $\ell' < \ell^*$ by this analysis. $\square$

We then provide a table to summarize the regret and communication cost of two proposed methods under different setting, respectively. Table 1 shows that for the regret upper bound of our proposed algorithm dealing with the known corruption setting, the corruption level $C$ serves as an additive factor in $\mathcal{O}(K \log T/\Delta^2 + KC/\Delta)$, which is independent of $T$. Besides, the communication cost of this algorithm is $\mathcal{O}(\log(\log T/\Delta^2 + KC/\Delta))$ level, which aligns with that of the original ETGS. As for the proposed Multi-layer ETGS race method handling unknown corruption, its regret upper

Table 1: Comparison of regret and communication cost for known $C$ and unknown $C$ settings

|  | Regret Bound | Communication Cost |
|---|---|---|
| **Known** $C$ | $\mathcal{O}\left(\frac{K\log T}{\Delta^2} + \frac{KC}{\Delta}\right)$ | $\mathcal{O}\left(\log\left(\frac{K\log T}{\Delta^2} + \frac{KC}{\Delta}\right)\right)$ |
| **Unknown** $C$ | $\mathcal{O}\left(K(\log T + C)\log^{1.5}T\left(\frac{1}{\Delta^2} + \frac{\sqrt{\log T}}{\Delta}\right)\right)$ | |

bound is around $\mathcal{O}(K\log^3 T/\Delta^2 + KC\log^2 T/\Delta^2)$. Compared with the known $C$ setting, the additional multiplicative $\log^2 T$ factor in the regret bound stems from running $\log T$ ETGS instances simultaneously and using at most $\log T$ rounds to communicate for achieving synchronization. Notably, in Multi-layer ETGS race for unknown corruptions, both communication cost and regret share the same order (due to constant sub-phase length $d$) while exhibiting sublinear scaling with respect to $T$, resulting in practically feasible communication overhead.

# 4 LOWER BOUND

In this section, we provide a regret lower bound for the bandit learning in decentralized matching markets with adversarial corruption. We mainly use the techniques in Sankararaman et al. (2021); Gupta et al. (2019) to prove this lower bound. The proof details are included in Appendix F.

Let $R_T(\boldsymbol{\nu}, \pi)$ denote the cumulative regret of a policy $\pi$ on the instance with arm distributions $\nu = \{\nu_{ij} : i \in [N], j \in [K]\}$ for a horizon of length $T$. Here, $\mathcal{P}$ denotes the set of all probability distributions with bounded support $[0, 1]$. This paper focuses on a class of policies termed uniformly consistent policies, defined as follows: A policy $\pi$ is called uniformly consistent if $\pi$ satisfies for all $\boldsymbol{\nu} \in \mathcal{P}$, all $\alpha \in (0, 1)$, the regret $\limsup_{T\to\infty} \frac{R_T(\boldsymbol{\nu}, \pi)}{T^\alpha} = 0$. This definition is used to eliminate tuning a policy to the current instance while admitting large regret in other instances. Before providing the lower bound, we first introduce a sub-class of bandits, where the stable matching is optimal. Specifically, we consider bandit instances where dominated arms are bad, i.e. for any instance $\boldsymbol{\nu}$ in this class, for all players $i \in [N]$, $\mu_{ij} < \mu_{ij_*^{(i)}}$ for all arms $j \in [K] \setminus \{j_*^{(i)}\}$. We call this class of instances Optimally Stable Bandits (OSB), as each agent is matched with its optimal arm in the stable matching. For OSB, another metric $\tilde{\Delta} := \min_{i \in [N], j \neq j_*^{(i)}} \mu_{i,j_*^{(i)}} - \mu_{i,j}$ is commonly used to characterize the problem's hardness.

**Theorem 4.1.** *For any player $i \in [N]$, under any decentralized universally consistent policy $\pi$, there exists an OSB bandit instance with Bernoulli rewards, where the regret of agent $i$ is lower bounded as*

$$Reg_i(T) \geq \Omega\left(\max\left\{(i-1)\left(\frac{\log T}{\tilde{\Delta}^2} + \frac{C}{K}\right), \frac{K\log T}{\tilde{\Delta}} + C\right\}\right). \tag{7}$$

**Remark 4.2.** *From Table 1, we know that the proposed robust variant of ETGS for the known corruption setting is near-optimal since the regret upper bound matches the lower bound, except for a slight difference in the definitions of the preference gaps $\Delta$ and $\tilde{\Delta}$. We claim that the mismatch between $\Delta$ and $\tilde{\Delta}$ is actually a fundamental open problem in the study of matching markets (even without corruption). It reflects the intrinsic difficulty and cost of achieving stable matchings through agent exploration and interaction. The best known lower bounds in the literature are established for the OSB setting and depend on the gap $\tilde{\Delta}$ (Sankararaman et al., 2021). For the unknown-corruption setting, we first note that a gap, steming from the different definitions of $\Delta$ and $\tilde{\Delta}$, still remains between the regret upper bound of the multi-layer ETGS race method and the lower bound. Moreover, our algorithm's upper bound incurs an additional multiplicative $\log^2 T$ factor compared to the lower bound. The multiplicative $\log^2 T$ factor arises because our method maintains $\log T$ ETGS instances to estimate the unknown $C$, and uses at most $\log T$ rounds for communication at the end of each sub-phase to achieve synchronization. These deteriorations in regret stem from the hardness of coordinating players in decentralized matching markets.*

# 5 EXPERIMENTS

In this section, we conduct simulation experiments to validate our theoretical findings. For the simulation setup, we set $N = 5$ and $K = 5$, and consider that the preference rankings are generated

as random permutations. The preference gap between any adjacent ranked arms is set as $0.2$. The feedback $X_{i,j}(t)$ for $p_i$ on $a_j$ at $t$ is drawn independently from the Gaussian distribution with mean $\mu_{i,j}$ and variance $1$. We adopt the corruption strategy proposed in Wang et al. (2024). We report the maximum cumulative player-optimal stable regret across all players. The results are averaged over ten independent runs with standard errors. We first compare our ETGS with widened confidence intervals and Multi-layer ETGS race with below baselines: Vanilla ETGS (Kong & Li, 2023), Phased ETC (Basu et al., 2021), and AETGS-E (Kong et al., 2024), all achieving player-optimal stable regret in decentralized matching markets. We evaluate all baselines under the corruption budget $C = 4000$ over $T = 500000$ rounds. As shown in Figure 1(a), two proposed algorithms incur the lowest cumulative regret among all baselines. We then investigate how varying corruption levels affect our algorithms, and set three corruption levels ($C \in \{1000, 2000, 4000\}$) to explore their effect on our algorithms. For this experiment, we set the sub-phase length $d = 1$ for Multi-layer ETGS race. Figure 1(b) shows that the regret of our algorithms increases with higher $C$. Besides, the algorithm with prior knowledge of $C$ outperforms the one without, under the same budget. Next, fixing $C = 1000$, we explore the influence of $d$ on Multi-layer ETGS race. The sub-phase length $d$ is set over $\{1, 10, 50\}$ to investigate its influence on Multi-layer ETGS race. Figure 1(c) depicts that large $d$ ($d = 10$ and $d = 50$) worsens the performance of Multi-layer ETGS race, consistent with our theoretical findings on the trade-off between communication overhead and learning efficiency. Finally, we present a thorough empirical analysis of the optimal choice of the hyperparameter $d$. We evaluate our method over $d \in \{1, 2, 4, 6, 8\}$. Recall that our theory suggests the optimal sub-phase length should satisfy $d = \mathcal{O}(\sqrt{\log T}) \approx 4.3$. In Figure 1(d), we observe that settings with $d = 4$ outperform the other choices, which aligns with our theoretical findings.

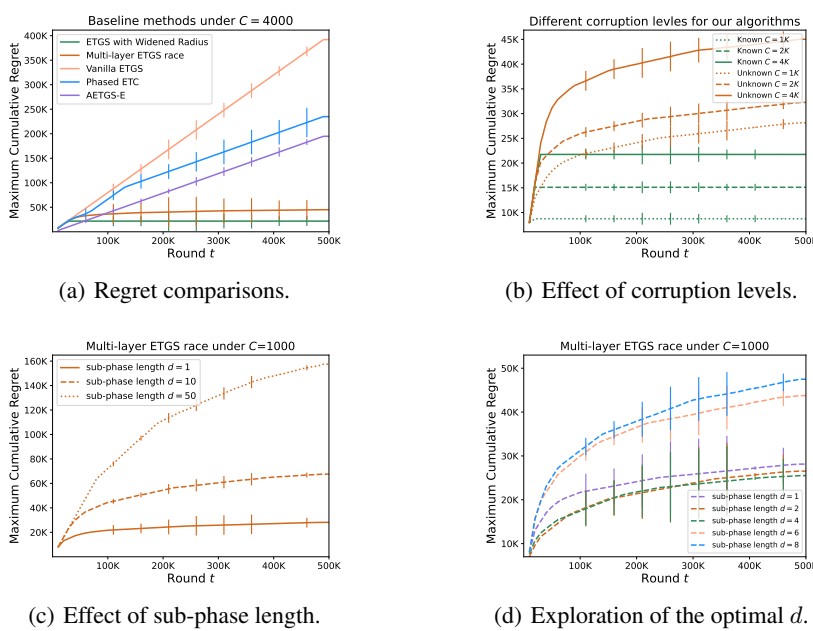

(a) Regret comparisons.

(b) Effect of corruption levels.

(c) Effect of sub-phase length.

(d) Exploration of the optimal $d$.

Figure 1: Experimental comparisons of baselines, and the effect of $C$ and $d$ on proposed methods.

## 6  CONCLUSION AND LIMITATIONS

This paper studies a novel bandit learning problem in decentralized matching markets under adversarial corruptions. We propose a robust ETGS variant to tackle known corruptions, and develop a Multi-layer ETGS race method to handle unknown corruptions. We derive regret upper bounds for both algorithms and also provide a lower bound to demonstrate their tightness. An important future direction is to design robust algorithms that achieve much higher communication efficiency in unknown corruption settings.

## ACKNOWLEDGEMENTS

The corresponding author Fang Kong is supported by National Natural Science Foundation of China (62506150) and Guangdong Basic and Applied Basic Research Foundation (2025A1515011412).

## ETHICS STATEMENT

This work is entirely theoretical and does not involve human subjects, personal data, or sensitive information. The research focuses on developing mathematical foundations and providing rigorous proofs for theoretical results in bandit learning. We do not foresee any direct ethical concerns, including issues of privacy, fairness, security, or legal compliance, arising from this work.

## REPRODUCIBILITY STATEMENT

All mathematical statements in our paper are fully specified: we clearly state all definitions, assumptions, lemmas and theorems, and provide complete proofs in the main text and appendix. In addition, for the simulations included in this paper, we provide detailed descriptions of the simulation setups. While our work does not depend on external datasets, the combination of theory and simulation documentation ensures that readers can fully reproduce and validate our results.

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

# APPENDIX

In the appendices, we provide more details and results omitted in the main paper. The appendices are structured as follows:

- Appendix A provides the related work part of this paper.

- Appendix B provides the missing details of the algorithmic procedures.

- Appendix C provides the proof of Theorem 3.1.

- Appendix D provides the proof of Lemma 3.3.

- Appendix E provides the proof of Theorem 3.4.

- Appendix F provides the proof of Theorem 4.1.

- Appendix G provides the LLM usage disclosure of this paper.

## A  RELATED WORK

Multi-armed bandits (MAB) have been extensively studied due to their broad applicability in sequential decision-making tasks (Slivkins, 2020; Lattimore & Szepesvári, 2020). Learning optimal stable matches for matching markets Gale & Shapley (1962); Li et al. (2025) with unknown preferences has become one of the important applications of MAB. Early work by Das & Kamenica (2005) introduced the bandit problem in matching markets, where multiple players and arms occupy opposing sides. Liu et al. (2020) later examined a variant with unknown one-sided preferences in general markets, deriving key theoretical guarantees. Further advancing this line of research, Liu et al. (2021); Kong et al. (2021) investigated decentralized matching markets where players act independently without central coordination. Sankararaman et al. (2021); Wang & Li (2024) consider scenarios in which participants' preferences adhere to specific assumptions to enhance learning efficiency. Kong & Li (2023) analyzed player-optimal stable regret in decentralized settings—a critical metric for practical applications. More recently, Kong et al. (2024) propose a novel algorithm and refined analysis to achieve an improved regret bound for this problem, and Kong et al. (2025); Lin et al. (2025) study the potential indifferent preferences in matching markets.

Another line of MAB research explores stochastic bandits with adversarial corruptions. For instance, Lykouris et al. (2018) proposes a randomized bandit algorithm robust to adversarial corruptions of stochastic rewards. Specifically, they consider a setting where the reward generated by each arm pull is stochastic but may be perturbed by an adversary before being revealed to the player. The result was subsequently improved by Gupta et al. (2019). Besides bandits robust to adversarial corruption, there is another line of works on best-of-both-worlds (BoBW) and best-of-three-worlds (BoTW) algorithms that aim to achieve near-optimal regret automatically adapting to stochastic, adversarial, and corrupted regimes. Early BoBW approaches design algorithms that perform competitively in both stochastic and adversarial settings by carefully combining exploration and exploitation mechanisms (Bubeck & Slivkins, 2012). Follow-up work further refines these ideas to obtain nearly optimal pseudo-regret guarantees across these regimes (Auer & Chiang, 2016; Zimmert & Seldin, 2019). In particular, Zimmert & Seldin (2019) propose an essentially optimal FTRL-based method with Tsallis-entropy regularization, which attains tight pseudo-regret bounds simultaneously in the stochastic and adversarial cases.

However, most existing work on stochastic bandits with adversarial corruptions focuses on the single-player scenario, with few studies addressing bandit learning in matching markets under corrupted feedback.

## B  DETAILS OF ALGORITHMIC PROCEDURES

In this section, we present the complete workflow of the vanilla ETGS algorithm Kong & Li (2023) and the monitoring subroutine used in our proposed algorithms. We then present the details of the leader selection and synchronization mechanism used in the Multi-layer ETGS race method.

## B.1 Workflow of Vanilla ETGS

First, we provided the algorithm procedure of the vanilla ETGS in Algorithm 3 to clarify the missing details of Phase 1 and Phase 3 used in our proposed Algorithm 1.

We emphasize that for our presented Multi-layer ETGS race method, designed to address unknown corruptions, its Phase 1 implementation is identical to that of the vanilla ETGS. Additionally, Phase 3 of the vanilla ETGS operates as a decentralized Gale-Shapley algorithm (Gale & Shapley, 1962). Consequently, the procedure of identifying the optimal stable matching in Algorithm 2 follows the same Phase 3 protocol as the vanilla ETGS.

Then we formally introduce the workflow of the vanilla ETGS. The first phase of ETGS proceeds in $N$ rounds (Line 3-8). At the first round $t = 1$, all players would propose to arm $a_1$ (Line 6) and only the player who is successfully accepted gets the index 1 (Line 8). In the second round, all of the other players (except for the player who gets index 1) still propose to $a_1$ (Line 6) and the only accepted player gets the index 2 (Line 8). Similar actions would be taken in the following rounds $3, 4, \ldots, N$. Intuitively, the index of each player $p_i$ is just the order of $p_i$ in the preference ranking of $a_1$. At the end of this phase, each player can obtain a distinct index. For the third phase (Line 24-28), it can be regarded as a decentralized Gale-Shapley algorithm (Gale & Shapley, 1962). During this phase, players aim to find and focus on the arm in the optimal stable matching with the estimated ranking $\sigma_i$. Specifically, $p_i$ would propose to arms one by one according to $\sigma_i$ until no rejection happens. When each player $p_i$'s estimated ranking $\sigma_i$ for the first $N$ arms is accurate, this procedure is expected to find the real optimal stable arm for each player.

---

**Algorithm 3** explore-then-Gale-Shapley (ETGS, from view of player $p_i$)

---

1: Input: player set $\mathcal{N}$, arm set $\mathcal{K}$, horizon $T$
2: Initialize: $\hat{\mu}_{i,j} = 0, T_{i,j} = 0, \forall j \in [K]$
3: //Phase 1, index estimation
4: Arm = $a_1$
5: **for** round $t = 1, 2, \ldots, N$ **do**
6:    $A_i(t) =$Arm
7:    **if** $\bar{A}_i(t) = A_i(t) = a_1$ **then**
8:       Index = $t$; Arm = $a_2$
9: //Phase 2, learn the preferences
10: **for** $\ell = 1, 2, \ldots$ **do**
11:    $F_\ell$ = False //whether the preference has been estimated well
12:    **for** $t = N + \sum_{\ell'=1}^{\ell-1}(2^{\ell'} + 1) + 1, \ldots, N + \sum_{\ell'=1}^{\ell-1}(2^{\ell'} + 1) + 2^\ell$ **do**
13:       $A_i(t) = a_{(\text{Index}+t-1)\%K+1}$
14:       Observe $X_{i,A_i(t)}(t)$ and update $\hat{\mu}_{i,A_i(t)}, T_{i,A_i(t)}$ if $\bar{A}_i(t) = A_i(t)$
15:    Compute $\text{UCB}_{i,j}$ and $\text{LCB}_{i,j}$ for each $j \in [K]$
16:    **if** $\exists \sigma$ such that $\text{LCB}_{i,\sigma_k} > \text{UCB}_{i,\sigma_{k+1}}$ for any $k \in [N]$ and $\text{LCB}_{i,\sigma_N} > \text{UCB}_{i,\sigma_k}$ for any $k = N + 1, N + 2, ..., K$ **then**
17:       $F_\ell$ = True and $\sigma_i = \sigma$
18:    Initialize $O_\ell = \emptyset$
19:    $t = N + \sum_{\ell'=1}^{\ell-1}(2^{\ell'} + 1) + 2^\ell + 1$
20:    $A_i(t) = a_{\text{Index}}$ if $F_\ell ==$ True and $A_i(t) = \emptyset$ otherwise
21:    Update $O_\ell = \cup_{i' \in [N]} \{\bar{A}_{i'}(t)\}$
22:    **if** $|O_\ell| == N$ **then**
23:       Enter in Phase 3 with $\sigma_i$; $t_2 = t$ //$t_2$ is the round when phase 2 ends
24: //Phase 3, find the optimal stable arm with $\sigma_i = (\sigma_{i,1}, \sigma_{i,2}, \ldots, \sigma_{i,K})$
25: Initialize $s = 1$
26: **for** $t = t_2 + 1, t_2 + 2, \ldots$ **do**
27:    $A_i(t) = a_{\sigma_{i,s}}$
28:    $s = s + 1$ if $\bar{A}_i(t) == \emptyset$

---

## B.2 The procedure of monitoring subroutine

In this subsection, we provide the procedure of the monitoring subroutine used in our algorithms in Subroutine 4.

---

**Subroutine 4** Monitoring

1: Input: $\{\text{UCB}_{i,j}, \text{LCB}_{i,j}\}_{j\in[K]}, t_k$
2: **if** $\exists \sigma$ such that $\text{LCB}_{i,\sigma_n} > \text{UCB}_{i,\sigma_{n+1}}$ for any $n \in [N]$ and $\text{LCB}_{i,\sigma_N} > \text{UCB}_{i,\sigma_n}$ for any $n = N+1, N+2, ..., K$ **then**
3: $\quad$ F$_k$ = True and $\sigma_i = \sigma$
4: Initialize $O_k = \emptyset$
5: $t = t_k$
6: $A_i(t) = a_{\text{Index}}$ if F$_k$ == True and $A_i(t) = \emptyset$ otherwise
7: Return $\sigma_i$ and $O_k = \cup_{i'\in[N]} \{\bar{A}_{i'}(t)\}$

---

Then we provide the motivation behind this monitoring subroutine. The core idea of this monitoring sub-routine is that, for each player $i \in [N]$, they use the feedback observed during the current sub-phase to compute $\text{UCB}_{i,k}$ and $\text{LCB}_{i,k}$ for all arms $k \in [K]$. The player then determines whether her preference ranking has been successfully estimated by checking whether the confidence intervals of all arms are mutually non-overlapping. After that, the player checks whether all other players have already estimated their respective preference ranks to determine whether to proceed to the next phase.

## B.3 The procedure of leader selection and synchronization mechanism

In the following, we introduce the details of leader selection and the synchronization mechanism, which are used in our proposed Multi-layer ETGS race method.

### B.3.1 Leader selection.

The procedure of leader selection is as follows: a leader is designated before the exploration phase begins, once each player has identified their own index. Specifically, at the end of Phase 1—when players obtain their respective indices—the protocol can simply assign Player 1 (i.e., the first player to acquire an index) as the leader.

**Reliability of leader selection.** As mentioned above, this protocol designates the player who first successfully acquires index 1 as the leader. All other players can observe the successful match between this player and arm 1, and they then recognize player 1 as the leader in the subsequent exploration phase. Based on the above procedure, there will always be a first player who successfully matches with arm 1, and thus there will always be a player selected as the leader. Therefore, it is difficult to disrupt this leader selection mechanism.

### B.3.2 Synchronization mechanism.

We observe that in a decentralized matching market, communication among players is realized through arm-proposing, where each player infers information by observing the arms pulled by others, not the direct communication among players (Kong & Li, 2023; Basu et al., 2021). Consequently, to implement the synchronization mechanism at the end of each sub-phase, the leader may spend up to $\log T$ additional rounds proposing a specific arm based on the result of layer sampling. By observing which arm the leader pulls, all other players can infer which layer (ETGS instance) they should enter in the next sub-phase.

## C Proof of Theorem 3.1

In the following, for convenience, let $\hat{\mu}_{i,j}(t), T_{i,j}(t), \text{UCB}_{i,j}(t), \text{LCB}_{i,j}(t)$ be the value of $\hat{\mu}_{i,j}, T_{i,j}, \text{UCB}_{i,j}$ and $\text{LCB}_{i,j}$ at the end of round $t$, respectively. Define $\mathcal{F} = \left\{\exists t \in [T], i \in [N], j \in [K] : |\hat{\mu}_{i,j}(t) - \mu_{i,j}| > \sqrt{\frac{6\log T}{T_{i,j}(t)}} + \frac{C}{T_{i,j}(t)}\right\}$ as the bad event that some pref-

erence is not estimated well during the horizon. Besides, we use the notation $\tilde{\mu}_{i,j}$ to denote the oracle empirical mean calculated by stochastic reward $r_{i,j}^{\mathcal{S}}(t)$ without any suffering corruption.

To provide the regret upper bound of the proposed Algorithm 1 for the known corruption setting, we first upper-bound the probability of the bad event $\mathcal{F}$ defined above.

**Lemma C.1.** *The upper bound for the probability of inaccurately estimating preferences is*

$$\mathbb{P}(\mathcal{F}) \leq 2NK/T. \tag{8}$$

*Proof.*

$$
\begin{aligned}
\mathbb{P}(\mathcal{F}) &\leq \mathbb{P}\left(\exists t, i, j : |\hat{\mu}_{i,j}(t) - \mu_{i,j}| > \sqrt{\frac{6 \log T}{T_{i,j}(t)}} + \frac{C}{T_{i,j}(t)}\right) \\
&\leq \mathbb{P}\left(\exists t, i, j : |\tilde{\mu}_{i,j}(t) - \mu_{i,j}| + |\hat{\mu}_{i,j}(t) - \tilde{\mu}_{i,j}| > \sqrt{\frac{6 \log T}{T_{i,j}(t)}} + \frac{C}{T_{i,j}(t)}\right) \\
&\leq \mathbb{P}\left(\exists t, i, j : |\tilde{\mu}_{i,j}(t) - \mu_{i,j}| > \sqrt{\frac{6 \log T}{T_{i,j}(t)}}\right) + \mathbb{P}\left(\exists t, i, j : |\hat{\mu}_{i,j}(t) - \tilde{\mu}_{i,j}| > \frac{C}{T_{i,j}(t)}\right) \\
&\overset{(a)}{=} \mathbb{P}\left(\exists t, i, j : |\tilde{\mu}_{i,j}(t) - \mu_{i,j}| > \sqrt{\frac{6 \log T}{T_{i,j}(t)}}\right) \\
&\leq \sum_t \sum_i \sum_j \sum_{s=1}^{t} \mathbb{P}\left(T_{i,j}(t) = s, |\tilde{\mu}_{i,j}(t) - \mu_{i,j}| > \sqrt{\frac{6 \log T}{s}}\right) \\
&\overset{(b)}{\leq} \sum_{t \in [T]} \sum_{i \in [N]} \sum_{j \in [K]} t \cdot 2 \exp(-3 \ln T) \\
&\leq 2NK/T,
\end{aligned}
\tag{9}
$$

where (a) holds due to the definition of $C$, and (b) holds since the Hoeffding's inequality. □

In the below Lemma C.2, we show that given $\neg\mathcal{F}$, once player $p_i$ observes that the UCB of an arm $a_j$ is smaller than the LCB of another arm $a_{j'}$, we are able to conclude that $p_i$ truly prefers $a_{j'}$ to $a_j$.

**Lemma C.2.** *Conditional on $\neg\mathcal{F}$, $UCB_{i,j}(t) < LCB_{i,j'}(t)$ implies $\mu_{i,j} < \mu_{i,j'}$.*

*Proof.* Given $\neg\mathcal{F}$, we know that $\forall t \in [T], i \in [N], j \in [K], |\tilde{\mu}_{i,j}(t) - \mu_{i,j}| \leq \sqrt{\frac{6 \log T}{T_{i,j}(t)}}$.

Then we have

$$
\begin{aligned}
\text{LCB}_{i,j}(t) &= \hat{\mu}_{i,j}(t) - \sqrt{\frac{6 \log T}{T_{i,j}(t)}} - \frac{C}{T_{i,j}(t)} \\
&\overset{(a)}{\leq} \tilde{\mu}_{i,j}(t) - \sqrt{\frac{6 \log T}{T_{i,j}(t)}} \\
&\leq \mu_{i,j} \\
&\leq \tilde{\mu}_{i,j} + \sqrt{\frac{6 \log T}{T_{i,j}(t)}} \\
&\overset{(b)}{\leq} \hat{\mu}_{i,j} + \sqrt{\frac{6 \log T}{T_{i,j}(t)}} + \frac{C}{T_{i,j}(t)} \\
&= \text{UCB}_{i,j}(t),
\end{aligned}
\tag{10}
$$

where (a) and (b) hold since the definition of $C$.

Thus, given $\neg\mathcal{F}$, we have $\mu_{i,j} \leq \text{UCB}_{i,j}(t) < \text{LCB}_{i,j'}(t) \leq \mu_{i,j'}$. □

The following Lemma C.3 provides an upper bound for the number of observations required to estimate the preference ranking for any player $p_i$ well.

**Lemma C.3.** *In round $t$, let $T_i(t) = \min_{j \in [K]} T_{i,j}(t)$ and $\bar{T}_i = \frac{384 \log T}{\Delta^2} + \frac{8C}{\Delta}$. Conditional on $\neg \mathcal{F}$, if $T_i(t) > \bar{T}_i$, we have $LCB_{i,\rho_{i,k}}(t) > UCB_{i,\rho_{i,k+1}}(t)$ for any $k \in [N]$, and $LCB_{i,\rho_{i,N}}(t) > UCB_{i,\rho_{i,k}}(t)$ for any $k = N+1, N+2, \dots, K$.*

*Proof.* By contradiction, suppose there exists $k \in [N]$ such that $\text{LCB}_{i,\rho_{i,k}}(t) \le \text{UCB}_{i,\rho_{i,k+1}}(t)$ or there exists $k = N+1, N+2, \dots, K$ such that $\text{LCB}_{i,\rho_{i,N}}(t) \le \text{UCB}_{i,\rho_{i,k}}(t)$. Without loss of generality, denote $j$ as the arm in the RHS and $j'$ as the arm in the LHS in above cases.

According to $\neg \mathcal{F}$ and the definition of LCB and UCB, we have

$$\mu_{i,j'} - 2\sqrt{\frac{6 \log T}{T_i(t)}} - \frac{2C}{T_i(t)} \le \text{LCB}_{i,j'}(t) \le \text{UCB}_{i,j}(t) \le \mu_{i,j} + 2\sqrt{\frac{6 \log T}{T_i(t)}} + \frac{2C}{T_i(t)}. \tag{11}$$

We can conclude that $\Delta_{i,j,j'} = \mu_{i,j'} - \mu_{i,j} \le 4\sqrt{\frac{6 \log T}{T_i(t)}} + \frac{4C}{T_i(t)}$.

If $T_i(t) > \frac{384 \log T}{\Delta^2} + \frac{8C}{\Delta}$, we have

$$
\begin{aligned}
\Delta_{i,j,j'} &\le 4\sqrt{\frac{6 \log T}{T_i(t)}} + \frac{4C}{T_i(t)} \\
&< \sqrt{\frac{96 \log T}{\frac{384 \log T}{\Delta^2} + \frac{8C}{\Delta}}} + \frac{4C}{\frac{384 \log T}{\Delta^2} + \frac{8C}{\Delta}} \\
&\le \frac{\Delta}{2} + \frac{\Delta}{2} \\
&= \Delta.
\end{aligned}
\tag{12}
$$

This implies that $T_i(t) \le \frac{384 \log T}{\Delta^2} + \frac{8C}{\Delta}$ and thus contradicts the fact that $T_i(t) > \bar{T}_i$. $\qquad \square$

According to the protocol of Algorithm 1, all players have the same observations, we can conclude that all of them would enter the third phase simultaneously. Denote $L_{\max}$ as the largest sub-phase number of Phase 2. That is to say, players enter in Phase 3 at the end of sub-phase $L_{\max}$.

Based on this observation and the lemmas provided above, we then provide the formal proof of Theorem 3.1 as follows.

*Proof of Theorem 3.1.* As introduced earlier, we denote $L_{\max}$ as the largest sub-phase number of Phase 2, i.e., the preference ranks are estimated well after this sub-phase.

The optimal stable regret for player $i$ can be bounded as follows,

$$
\begin{aligned}
Reg_i(T) &= \mathbb{E}\Big[ \sum_t (\mu_{i,m_i^*} - X_i(t)) \Big] \\
&\le \mathbb{E}\Big[ \sum_t \mathbb{I}\{\bar{A}(t) \ne m^*\} \cdot \Delta_{i,\max} \Big] \\
&\le N\Delta_{i,\max} + \mathbb{E}\Big[ \sum_{t=N+1}^{T} \mathbb{I}\{\bar{A}(t) \ne m^*\} | \neg \mathcal{F} \Big] \cdot \Delta_{i,\max} + \mathbb{P}(\mathcal{F}) \cdot T \cdot \Delta_{i,\max} \\
&\le N\Delta_{i,\max} + \mathbb{E}\Big[ \sum_{k=1}^{L_{\max}} (d_k + 1) + N^2 | \neg \mathcal{F} \Big] \cdot \Delta_{i,\max} + \mathbb{P}(\mathcal{F}) \cdot T \cdot \Delta_{i,\max} \\
&\overset{(a)}{\le} N\Delta_{i,\max} + \mathbb{E}\Big[ \sum_{k=1}^{L_{\max}} (d_k + 1) + N^2 | \neg \mathcal{F} \Big] \cdot \Delta_{i,\max} + 2NK \cdot \Delta_{i,\max},
\end{aligned}
\tag{13}
$$

where (a) comes from Lemma C.1.

Based on Lemma C.3, we know that Phase 2 proceeds in at most $L_{\max}$ sub-phases where

$$L_{\max} = \min\Big\{k : \sum_{k'=1}^{k} d_k \geq 8K\Big(\frac{48\log T}{\Delta^2} + \frac{C}{\Delta}\Big)\Big\}. \tag{14}$$

Recall that we select $d_k = 2^k$ and set the confidence radius as $\sqrt{\frac{6\log T}{T_{i,j}}} + \frac{C}{T_{i,j}}$ in Algorithm 1, based on the definition of $L_{\max}$, we have

$$\sum_{k'=1}^{L_{\max}} 2^{k'} \leq 16K\Big(\frac{48\log T}{\Delta^2} + \frac{C}{\Delta}\Big). \tag{15}$$

Hence we have $L_{\max} = \log\Big(16K\big(\frac{48\log T}{\Delta^2} + \frac{C}{\Delta}\big)\Big)$, and the regret can be bounded as follows

$$\begin{aligned}
Reg_i(T) \leq{}& \Big(16K\Big(\frac{48\log T}{\Delta^2} + \frac{C}{\Delta}\Big) + \log\Big(16K\big(\frac{48\log T}{\Delta^2} + \frac{C}{\Delta}\big)\Big)\Big) \cdot \Delta_{i,\max} \\
& + N\Delta_{i,\max} + N^2\Delta_{i,\max} + 2NK\Delta_{i,\max}.
\end{aligned} \tag{16}$$

$\square$

## D  PROOF OF LEMMA 3.3

In this section, we provide the proof of Lemma 3.3. We bound with high probability the total corruption suffered by the ETGS instance with sampling probability $1/C$. Similar to Lykouris et al. (2018), we also use a Bernstein-style martingale concentration inequality.

**Lemma D.1** (Lemma 1 in Beygelzimer et al. (2011)). *Let $X_1, \ldots, X_T$ be a sequence of real-valued random numbers. Assume, for all $t$, that $X_t \leq R$ and that $\mathbb{E}[X_t|X_1, \ldots, X_{t-1}] = 0$. Also let*

$$V = \sum_{t=1}^{T} \mathbb{E}[X_t^2|X_1, \ldots, X_{t-1}].$$

*Then, for any $\delta > 0$:*

$$\mathbb{P}\left[\sum_{t=1}^{T} X_t > R\ln(1/\delta) + \frac{e-2}{R}\cdot V\right] \leq \delta.$$

*Proof of Lemma 3.3.* Let $Z_{i,j}(t)$ be the corruption that is observed in the $t$-th round of the ETGS instance with sampling probability $P = 1/C$ for the match between the player $p_i$ and arm $a_j$. For every round $t$, $C_{i,j}(t)$ is the corruption selected by the adversary for the match of $p_i$ and $a_j$. $T_k$ denotes the $k$-th sub-phase. We define $Z_i(k)$ as the total corruption observed in the $k$-th sub-phase of the ETGS instance with sampling probability $P = 1/C$ for player $i$. In words, $Z_i(k)$ is defined as $Z_i(k) := \sum_{t \in T_k} Z_{i,A_i(t)}(t)$. According to our algorithm, $Z_i(k)$ is a Bernoulli random variable: $Z_i(k) = C_i(k)$ with the probability $P$ and $Z_i(k) = 0$ with the probability $1 - P$, where $C_i(k) := \sum_{t \in T_k} C_{i,A_i(t)}(t)$.

Then we define the martingale sequence as

$$X_i(k) := Z_i(k) - \mathbb{E}\Big[Z_i(k)|\mathcal{H}(1:k-1)\Big], \tag{17}$$

where $\mathcal{H}(1:k-1)$ corresponds to the history up to sub-phase $k$.

Note that

$$\begin{aligned}
\mathbb{E}\Big[(X_i(k))^2|\mathcal{H}(1:k-1)\Big] &= P(C_i(k) - PC_i(k))^2 + (1-P)(PC_i(k))^2 \\
&= P(1-P)^2(C_i(k))^2 + (1-P)(PC_i(k))^2 \\
&= P(1-P)(C_i(k))^2(P + (1-P)) \\
&= P(1-P)(C_i(k))^2.
\end{aligned} \tag{18}$$

Since we set the length of each sub-phase as a constant $d$, we can bound the term $(C_i(k))^2$ following

$$(C_i(k))^2 = \left( \sum_{t \in T_k} C_i(t) \cdot 1 \right)^2 \leq \sum_{t \in T_k} (C_i(t))^2 \sum_{t \in T_k} 1^2 \leq d \sum_{t \in T_k} (C_i(t))^2 \leq d \sum_{t \in T_k} C_i(t), \quad (19)$$

where the first inequality holds since Cauchy-Schwarz inequality and the last inequality holds due to $C_i(t) \in [0, 1]$.

Therefore, summing over all the sub-phases, the variance $V$ becomes

$$V = \sum_k \mathbb{E}\Big[(X_i(k))^2 | \mathcal{H}(1 : k - 1)\Big]$$
$$\leq P \sum_k \Big(C_i(k)\Big)^2$$
$$\leq Pd \sum_k \sum_{t \in T_k} C_i(t) \quad (20)$$
$$= Pd \sum_{t \in [T]} C_i(t)$$
$$\leq d,$$

where the last step holds due to the definition of $C$ and $P = \frac{1}{C}$.

Then we turn to upper-bound $X_i(k)$ as follows,

$$X_i(k) = Z_i(k) - \mathbb{E}\Big[Z_i(k) | \mathcal{H}(1 : k - 1)\Big]$$
$$\leq \sum_{t \in T_k} Z_i(t) \quad (21)$$
$$\leq d,$$

where the last inequality holds since the rewards are in $[0, 1]$ (thus the instant corruption for each match in one round should be in $[0, 1]$).

Applying Lemma D.1, we show that, w.p. $1 - \frac{1}{T}$:

$$\sum_k X_i(k) \leq d \log(T) + \frac{(e - 2)V}{d}$$
$$\leq d \log(T) + \frac{d}{d} \quad (22)$$
$$= d \log(T) + 1$$

We also know that

$$\mathbb{E}\Big[\sum_k Z_i(k) | \mathcal{H}(1 : k - 1)\Big] = \sum_k \mathbb{E}\Big[Z_i(k) | \mathcal{H}(1 : k - 1)\Big]$$
$$= P \sum_k C_i(k)$$
$$= \frac{1}{C} \sum_{t=1}^T C_{i,A_i(t)}(t) \quad (23)$$
$$\leq 1.$$

To sum up, w.p. $1 - \frac{1}{T}$, the total corruption incurred by the ETGS instance with $P = \frac{1}{C}$ for player $i$ is

$$\sum_t Z_{i,A_i(t)}(t) = \sum_k X_i(k) + \mathbb{E}\Big[\sum_k Z_i(k) | \mathcal{H}(1 : k - 1)\Big]$$
$$\leq d \log(T) + 2. \quad (24)$$

$\square$

# E   PROOF OF THEOREM 3.4

For each layer $\ell \in [\log T]$, we denote $\hat{\mu}_{i,j}^{\ell}(t), T_{i,j}^{\ell}(t), \mathrm{UCB}_{i,j}^{\ell}(t), \mathrm{LCB}_{i,j}^{\ell}(t)$ as the value of $\hat{\mu}_{i,j}^{\ell}, T_{i,j}^{\ell}, \mathrm{UCB}_{i,j}^{\ell}$ and $\mathrm{LCB}_{i,j}^{\ell}$ at the end of round $t$, respectively. In the following, we define $\mathcal{F}_S^{\ell} = \left\{ \exists t \in [T], i \in [N], j \in [K] : \left| \hat{\mu}_{i,j}^{\ell}(t) - \mu_{i,j} \right| > \sqrt{\frac{6 \log T}{T_{i,j}^{\ell}(t)}} + \frac{2d \log T}{T_{i,j}^{\ell}(t)} \right\}$ as the event that some preference is not estimated well during the horizon for the $\ell$-th layer. Besides, we denote $\mathcal{F}_C$ as the event that there exists one ETGS instance of those layers $\ell', \ell' \in [\log T]$ whose sampling probability satisfying $2^{-\ell'} \leq 1/C$, the actual experienced total corruption of some player in this instance is larger than $d \log T + 2$. And we define the bad event $\mathcal{F} = \left( \cup_{\ell \in [\log T]} \mathcal{F}_S^{\ell} \right) \cup \mathcal{F}_C$.

In the following, we also give an upper bound for the probability of the bad event $\mathcal{F}$.

**Lemma E.1.** *The upper bound for the probability of the bad event $\mathcal{F}$ is*

$$\mathbb{P}(\mathcal{F}) \leq \frac{2NK \log T}{T} + \frac{N \log T}{T}. \tag{25}$$

*Proof.* We first define the notation $C_i^{\ell}(T)$ as the actual total corruption experienced by the $\ell$-th ETGS instance of player $i$. Recall that we select the constant $d$ satisfying $d \log T + 2 \leq 2d \log T$, we can thus upper-bound the probability of the bad event $\mathcal{F}$ as follows

$$\begin{aligned}
\mathbb{P}(\mathcal{F}) &\leq \sum_{\ell \in [\log T]} \mathbb{P}(\mathcal{F}_S^{\ell}) + \mathbb{P}(\mathcal{F}_C) \\
&\overset{(a)}{\leq} \sum_{\ell \in [\log T]} \mathbb{P}\left( \exists t, i, j : |\hat{\mu}_{i,j}^{\ell}(t) - \mu_{i,j}| > \sqrt{\frac{6 \log T}{T_{i,j}^{\ell}(t)}} + \frac{2d \log T}{T_{i,j}^{\ell}(t)} \right) \\
&\quad + \sum_{\ell' : 2^{-\ell'} \leq 1/C} \mathbb{P}(\exists i : C_i^{\ell'}(T) > 2d \log(T)) \\
&\overset{(b)}{\leq} \sum_{\ell \in [\log T]} \sum_{t \in [T]} \sum_{i \in [N]} \sum_{j \in [K]} t \cdot 2 \exp(-3 \ln T) + \sum_{i \in [N]} \frac{\log T}{T} \\
&\leq \frac{2NK \log T}{T} + \frac{N \log T}{T},
\end{aligned} \tag{26}$$

where (a) holds due to our selection of enlarged radius for the unknown $C$ setting, and (b) holds based on the similar analysis in the proof of Lemma C.1 and Lemma 3.3. $\qquad \square$

Based on the results of Lemma 3.3, we know that the actual corruption suffered by the layers $\ell$ satisfies that $2^{\ell} \geq C, \ell \in [\log T]$ is at most $d \log T + 2$. Since we select the constant $d$ satisfying $d \log T + 2 \leq 2d \log T$ and set the confidence interval as $\sqrt{\frac{6 \log T}{T_{i,j}^{\ell}}} + \frac{2d \log T}{T_{i,j}^{\ell}}$, we can immediately have the following results.

**Lemma E.2.** *In round $t$, let $T_i^{\ell}(t) = \min_{j \in [K]} T_{i,j}^{\ell}(t)$ and $\bar{T}_i^{\ell} = \frac{384 \log T}{\Delta^2} + \frac{2d \log T}{\Delta}$. Conditional on $\neg \mathcal{F}$, if $T_i^{\ell}(t) > \bar{T}_i^{\ell}$, we have $LCB_{i,\rho_{i,k}}^{\ell}(t) > UCB_{i,\rho_{i,k+1}}^{\ell}(t)$ for any $k \in [N]$, and $LCB_{i,\rho_{i,N}}^{\ell}(t) > UCB_{i,\rho_{i,k}}^{\ell}(t)$ for any $k = N+1, N+2, \ldots, K$.*

Below, we present the proof of Theorem 3.4.

*Proof of Theorem 3.4.* The optimal stable regret for player $i$ can be bounded as follows,

$$
\begin{aligned}
Reg_i(T) &= \mathbb{E}\Big[ \sum_t (\mu_{i,m_i^*} - X_i(t)) \Big] \\
&\leq \mathbb{E}\Big[ \sum_t \mathbb{I}\{\bar{A}(t) \neq m^*\} \cdot \Delta_{i,\max} \Big] \\
&\leq N\Delta_{i,\max} + \mathbb{E}\Big[ \sum_{t=N+1}^{T} \mathbb{I}\{\bar{A}(t) \neq m^*\}|\neg\mathcal{F} \Big] \cdot \Delta_{i,\max} + \mathbb{P}(\mathcal{F}) \cdot T \cdot \Delta_{i,\max} \\
&\leq (N + 2NK\log T + N\log T)\Delta_{i,\max} + \mathbb{E}\Big[ \sum_{t=N+1}^{T} \mathbb{I}\{\bar{A}(t) \neq m^*\}|\neg\mathcal{F} \Big] \cdot \Delta_{i,\max},
\end{aligned}
\tag{27}
$$

where the last inequality comes from Lemma E.1

In the following, we focus on the regret contributed by those layers whose suffered actual corruption is smaller than $C$, and the regret contributed by the layers that are not tolerant to the corruption $C$, respectively. We first define the minimum layer that is robust to corruption: $\ell^* := \arg\min_\ell [2^\ell > C]$. Hence, for the robust layers $\ell' > \ell^*$, according to Lemma 3.3 and Lemma E.2, we can establish a regret upper bound of $16K(\frac{24\log T}{\Delta^2} + \frac{d\log T}{\Delta}) \cdot \Delta_{i,\max}$. Since there are at most $\log T$ layers, the total regret coming from these robust layers is $16K\log T(\frac{24\log T}{\Delta^2} + \frac{d\log T}{\Delta}) \cdot \Delta_{i,\max}$.

For the layers $\ell < \ell^*$ that are not tolerant to the corruption, i.e., $2^\ell < C$, we know the optimal stable matches of these faster layers will be modified to keep the same as that of $\ell^*$ when $\ell^*$ has estimated its own preference rank well. Hence, in expectation, this is at most $C \cdot T_{i,j}^{\ell^*}$ times as every move in the layer $\ell^*$ occurs with probability $1/C$ of these moves are matches of arm $j$ while it is the optimal stable matching for the faster algorithms. Formally, we define the number of times that the optimal layer is selected as $T_{i,j}^{\ell^*}$. In expectation, we know the layers $\ell < \ell^*$ have finished the estimation of their preference ranks before $\ell^*$. Thus, we have

$$
\mathbb{E}[\tilde{T}_{i,j}] = \frac{1}{P_{\ell^*}} T_{i,j}^{\ell^*} - T_{i,j}^{\ell^*} = C T_{i,j}^{\ell^*} - T_{i,j}^{\ell^*} \leq C T_{i,j}^{\ell^*},
\tag{28}
$$

where $\tilde{T}_{i,j}$ denotes the number of times that layers $\ell < \ell^*$ are chosen.

Based on the above analysis, the total regret contributed by the faster layers is $16KC(\frac{24\log T}{\Delta^2} + \frac{d\log T}{\Delta}) \cdot \Delta_{i,\max}$.

According to the protocol of our algorithm, the number of sub-phases in Phase 2 is at most $\lfloor 16K(\frac{24\log^2 T}{d\Delta^2} + \frac{\log^2 T}{\Delta} + \frac{24C\log T}{d\Delta^2} + \frac{C\log T}{\Delta}) \rfloor$ and thus the total communication cost is $\lfloor 16K\log T(\frac{24\log^2 T}{d\Delta^2} + \frac{\log^2 T}{\Delta} + \frac{24C\log T}{d\Delta^2} + \frac{C\log T}{\Delta}) \rfloor$.

To sum up, the regret upper bound of Multi-layer ETGS is

$$
\begin{aligned}
&Reg_i(T) \\
&\leq (N + N^2\log T + 2NK\log T + N\log T)\Delta_{i,\max} + 16K\log T\left(\frac{24\log T}{\Delta^2} + \frac{d\log T}{\Delta}\right) \cdot \Delta_{i,\max} \\
&\quad + 16KC\left(\frac{24\log T}{\Delta^2} + \frac{d\log T}{\Delta}\right) \cdot \Delta_{i,\max} \\
&\quad + 16K\log T\left(\frac{24\log^2 T}{d\Delta^2} + \frac{\log^2 T}{\Delta} + \frac{24C\log T}{d\Delta^2} + \frac{C\log T}{\Delta}\right) \cdot \Delta_{i,\max}. \\
&= \mathcal{O}\left(\left(\frac{Kd\log T(\log T + C)}{\Delta} + \frac{K\log^2 T(\log T + C)}{d\Delta^2} + \frac{K\log^2 T(\log T + C)}{\Delta}\right) \cdot \Delta_{i,\max}\right).
\end{aligned}
\tag{29}
$$

$\square$

# F    PROOF OF THEOREM 4.1

In this section, we provide the proof of Theorem 4.1 by utilizing the main results from Sankararaman et al. (2021); Gupta et al. (2019).

Before providing the formal proof, we provide a vital lemma proposed in Sankararaman et al. (2021).

**Lemma F.1.** *Under any decentralized universally consistent algorithm $\pi$, there exist a OSB bandit instance with Bernoulli rewards, where the regret of player $i \in [N]$ is lower bounded as $\Omega\left(\max\left\{\frac{(i-1)\log T}{\tilde{\Delta}^2}, \frac{K\log T}{\tilde{\Delta}}\right\}\right)$, where $\tilde{\Delta} := \min_{i\in[N],j\neq j_*^{(i)}} \mu_{i,j_*^{(i)}} - \mu_{i,j}$.*

*Proof of Theorem 4.1.* Recall that we define $j_*^{(i)} := \arg\max_{j\in[K]} \mu_{ij}$ recursively and we denote $\tilde{\Delta}_j^i := \mu_{i,j_*^{(i)}} - \mu_{i,j}$ and $\tilde{\Delta}_{\min}^i := \arg\min_{j\in[K]} \mu_{i,j_*^{(i)}} - \mu_{i,j}$ in the following.

For any player $i \in [N]$, we know its pseudo regret under policy $\pi$ and any bandit instance $\boldsymbol{\nu}$ is

$$R_T^i(\boldsymbol{\nu}, \pi) = \sum_{j\in[K]} \tilde{\Delta}_j^i \mathbb{E}_{\nu,\pi}[N_j^i(T)] + \mu_{i,j_*^{(i)}}[B^i(T)], \tag{30}$$

where $B^i(t)$ denotes the number of time the agent $i$ is blocked up to time $t$.

We claim that this is true as for each matching conflict the player $p_i$ obtains $\mu_{ij_*^i}$ regret (0 reward) in expectation, and for each successful match of arm $j$ it obtains $\tilde{\Delta}_j^i$ regret.

Based on it, a trivial regret lower bound is

$$R_T^i(\boldsymbol{\nu}, \pi) \geq \sum_{j\in[K]} \tilde{\Delta}_j^i \mathbb{E}_{\nu,\pi}[N_j^i(T)]. \tag{31}$$

To illustrate the effect of corruption on the problem hardness, below we follow the basic idea of Gupta et al. (2019) to conduct a finer-grained analysis for Eq. (31). In specific, for any player $i \in [N]$, we consider another bandit instance $\nu$: the deterministic reward of the optimal arm is $\tilde{\Delta}$ and the reward of each sub-optimal arm is zero without any noise. Given that the regret defined in this paper is based on an adaptive adversary who can inject corruptions based on the historical information, we can thus consider a weaker adversary here to provide a valid lower bound of regret. In specific, this weaker adversary swaps the rewards of matched arms with probability $1/2$ at each round. Thus, in expectation, there are at most a total amount of $\lfloor C/\tilde{\Delta} \rfloor$ rounds where the rewards are swapped during the first $\lfloor 2C/\tilde{\Delta} \rfloor$ rounds. This makes the arms appear indistinguishable to the algorithm during these rounds.

Besides, we know that in expectation the total amount of rounds where the reward of arm $j \in [K]$ is corrupted is roughly $\lfloor \frac{C}{K\tilde{\Delta}} \rfloor$. Then we have $\mathbb{E}'[N_j^i] \geq \lfloor \frac{C}{K\tilde{\Delta}} \rfloor$, where $\mathbb{E}'$ denotes the expectation operator taken over the randomness from the attacking strategy of the considered adversary.

Based on the above analysis, we have

$$\begin{aligned} R_T^i(\boldsymbol{\nu}, \pi) &\geq \sum_{j\in[K]} \tilde{\Delta}_j^i \mathbb{E}_{\nu,\pi}[N_j^i(T)] \\ &\geq \sum_{j\neq j_*^i} \tilde{\Delta} \mathbb{E}'_{\nu,\pi}[N_j^i(T)] \\ &= (K-1)\tilde{\Delta} \lfloor \frac{C}{K\tilde{\Delta}} \rfloor \\ &= \Omega(C), \end{aligned} \tag{32}$$

where the second inequality holds since we consider a weaker adversary here instead of the adaptive adversary using the historical information to maximize the pseudo regret.

Applying Lemma F.1, we know that

$$\liminf_{T\to\infty} \frac{R_T^i(\boldsymbol{\nu})}{\log T} \geq \frac{K}{2\tilde{\Delta}}. \tag{33}$$

Based on the analysis of these two instances, we have

$$R_T^i(\boldsymbol{\nu}) \gtrsim \max\left\{\frac{K\log T}{\tilde{\Delta}}, C\right\} \geq \Omega\left\{\frac{K\log T}{\tilde{\Delta}} + C\right\}. \tag{34}$$

Referring to similar proof proposed in Sankararaman et al. (2021), we know that for an OSB instance, the number of times the players 1 to $(i-1)$ matches arm $j_*^i$ successfully, the player $p_i$ should either move to a sub-optimal arm or it is blocked. In the best possible scenario, $p_i$ successfully matches its second best arm, in each of these instances. This holds as $\tilde{\Delta}_{\min}^i \leq \mu_{ij_*^i}$ for non-negative rewards. Therefore, the regret from the events when players $p_1$ to $p_{i-1}$ matches arm $j_*^i$ successfully, is lower bounded following

$$\begin{aligned}
R_T^i(\boldsymbol{\nu}, \pi) &\geq \mu_{i,j_*^{(i)}}[B^i(T)] \\
&\geq \sum_{i'=1}^{i-1} \tilde{\Delta}_{\min}^i \mathbb{E}_{\nu,\pi}[N_{j_*^i}^{i'}] \\
&\geq \sum_{i'=1}^{i-1} \tilde{\Delta}\mathbb{E}_{\nu,\pi}'[N_{j_*^i}^{i'}] \\
&\geq \sum_{i'=1}^{i-1} \tilde{\Delta}\lfloor\frac{C}{K\tilde{\Delta}}\rfloor \\
&= \Omega\Big(\sum_{i'=1}^{i-1}\frac{C}{K}\Big).
\end{aligned} \tag{35}$$

Similarly, applying Lemma F.1 and then we have

$$\liminf_{T\to\infty}\frac{R_T^i(\boldsymbol{\nu})}{\log T} \geq \frac{i-1}{\tilde{\Delta}^2}. \tag{36}$$

To sum up, we have

$$R_T^i(\boldsymbol{\nu}) \geq \Omega\left\{(i-1)\big(\frac{\log T}{\tilde{\Delta}^2} + \frac{C}{K}\big)\right\}. \tag{37}$$

Combining the results in Eq. (33) and Eq. (37), we can roughly lower-bound the pseudo regret by

$$Reg_i(T) = \Omega\Big(\max\Big\{(i-1)\big(\frac{\log T}{\tilde{\Delta}^2} + \frac{C}{K}\big), \frac{K\log T}{\tilde{\Delta}} + C\Big\}\Big). \tag{38}$$

$\square$

## G  THE USE OF LARGE LANGUAGE MODELS (LLMS)

We used a large language model (LLM) only for editing—correcting grammar, spelling, word choice, and overall phrasing to improve readability. The LLM did not contribute to research ideation, methodology, experiments, data analysis, results, proofs, or theoretical claims. All such components were conceived, developed, and validated solely by the authors. We assume full responsibility for the final manuscript, including parts influenced by the LLM, and declare that no content generated via the LLM constitutes plagiarism or scientific misconduct.

