# OpenReview forum: "Bandit Learning in Matching Markets Robust to Adversarial Corruptions"
_ICLR.cc/2026/Conference — ICLR 2026 Poster_

### Official Review · Reviewer_wBB7 · 2025-10-29

**Soundness:** 3
**Presentation:** 3
**Contribution:** 3
**Rating:** 6
**Confidence:** 3

**Summary:**

This paper studies bandit learning for two-sided decentralized matching markets with corrupted feedback, i.e., the feedback revealed to the learner could be corrupted from the ground truth within some certain corruption level $C$. Algorithms are presented for both known $C$ and unknown $C$ setting.

In the known $C$ setting, one can simply enlarge the confidence interval so that the ground truth still lies in it (w.h.p). However, it is not applicable to unknown $C$ case. Therefore, in the later setting, in the algorithm design, multiple algorithm instances are created, each of which are tailored to a specific corruption level and are sample with different probability. Eventually, one can show theoretical guarantee when coordinating all these instances together.

**Strengths:**

1. A new setup that has not been addressed.
2. Theoretical upper and lower bounds.

**Weaknesses:**

n/a. please see questions.

**Questions:**

1. In terms of how to address unknown corruption level, how would the design in this paper different from Lykouris et al. (2018)?
2. Since each instance is sampled with a fixed probability, how can one adapt to the best/correct one on the fly? (or do we need to?)
3. In Table 1, why are the regret bound and comm. cost exactly the same in the unknown $C$ setting?
4. From the lower bounds in Thm. 4.1, how can we claim Remark 4.2? The suboptimality gap has a different definition, and the unknown $C$ upper bound looks quite complicated, can the authors please give a breakdown?

---

> ### Author Response · Authors · 2025-11-20
> **Response to Reviewer wBB7**
>
> We sincerely appreciate the reviewer’s valuable comments. Below we provide point-by-point responses to each comment.
>
>
> >Q1: Difference with Lykouris et al. (2018).
>
> R: We clarify the conceptual and technical differences between our work and Lykouris et al. (2018) in the following. A direct application of their vanilla MLAER algorithm to decentralized matching markets is not feasible, as it would inevitably lead to frequent conflicts among players. In their single-player setting, a single player internally maintains and schedules all MLAER layers, ensuring that only one instance proposes an action at any time. In contrast, in our multi-player environment, if each player independently samples a layer, two or more players may propose to the same arm in the same round, creating unavoidable interference and preventing the system from reaching stable matchings.
>
> To address this structural challenge, we introduce a synchronization mechanism that is fundamentally absent in the single-player framework. We partition the learning process into subphases of length $d$ and enforce that all players operate within the same ETGS layer during each subphase, following a consistent round-robin schedule. This coordination is achieved by allowing a designated leader to encode and transmit the next-layer index to all other players through carefully constructed arm proposals.
>
> Moreover, this multi-player structure requires analytical tools that go beyond those in Lykouris et al. (2018). Their analysis is conducted at the level of individual rounds, while ours develops a new sub-phase level martingale concentration inequality that yields confidence bounds specifically tailored to the decentralized matching setting. This refinement is essential because corruption accumulates over subphases rather than individual rounds, and because synchronization constraints fundamentally alter the stochastic process.
>
> Finally, the synchronization mechanism together with the sub-phase level concentration analysis reveals a new tradeoff: the choice of $d$ mediates a tension between communication and robustness. Smaller $d$ reduces exploration cost but increases communication overhead, while larger $d$ reduces communication but amplifies exploration cost within each ETGS layer. Managing this tradeoff is intrinsic to decentralized matching markets and highlights the substantial conceptual departure from the single-player MLAER framework.
>
>
>
>
> >Q2: Adaptation to the best/correct layer on the fly.
>
> R: The design of the sampling probabilities and the layer structure already ensures automatic adaptation to the optimal layer $\ell^{ * }$.  Below, we explain how the regret contributed by the non-optimal layers is effectively controlled.
>
> 1. We first analyze the regret mainly contributed by the "faster layers", i.e., those layers whose sampling probability greater than $1/C$. Next we check the number of exploration rounds unitl $\ell^{ * }$ finishes estimating its stable matching.  Based on our derived sub-phase martingale concentration inequality, we know that  the layer $\ell^{ * }$ needs to explore for $\mathcal{O}\left(K\left(\frac{24\log T}{\Delta^2}+\frac{d\log T}{\Delta} \right)\right)$ rounds to estimate the correct player-optimal stable matchings with high probability. Given that the sampling probability for the optimal layer is $1/C$, the total exploration rounds until  the optimal layer has estimated its stable mathcing is $\mathcal{O}\left(KC\left(\frac{24\log T}{\Delta^2}+\frac{d\log T}{\Delta} \right)\right)$ in expectation. Thus the suffered regret is  $\mathcal{O}\left(KC\left(\frac{24\log T}{\Delta^2}+\frac{d\log T}{\Delta} \right)\cdot \Delta_{i,\max}\right)$ corresponding to these exploration rounds.
>
>
> 2. We then analyze the "slower layers", whose sampling probability is less than $1/C$. For any slower layer $\ell$ such that $2^{-\ell} < 1/C$, our sub‑phase martingale concentration bound shows that it suffers at most $d\log T + 2$ corruption (with high probability). Accordingly, the regret suffered by each such slower layer is also $\mathcal{O}\left(K(\tfrac{24\log T}{\Delta^2} + \tfrac{d\log T}{\Delta})\cdot \Delta_{i,\max}\right)$. Since there are at most $\log T$ of these slower layers, their total contributed regret is $\mathcal{O}\left(K \log T\cdot (\frac{24\log T}{\Delta^2} + \frac{d\log T}{\Delta})\cdot \Delta_{i,\max}\right)$.
>
> Combining both parts, the total regret from all non-optimal layers $\ell, \ell \neq \ell^*$ is about $\mathcal{O}\left(KC\left(\frac{24\log T}{\Delta^2}+\frac{d\log T}{\Delta} \right)\cdot \Delta_{i,\max}+K\log T \left(\frac{24\log T}{\Delta^2}+\frac{d\log T}{\Delta} \right)\cdot \Delta_{i,\max}\right)$, which grows sublinearly in $T$.  Therefore, we conclude that our algorithm effectively controls the regret contributed by all ETGS layers.

---

> ### Author Response · Authors · 2025-11-20
> **Response to Reviewer wBB7**
>
> We sincerely appreciate the reviewer’s valuable comments. Below we provide point-by-point responses to each comment.
>
> > Q3: Discussion on the Regret Bound vs. Communication Cost under Unknown $C$.
>
> R: First, we clarify that the regret bound and the communication cost in the unknown corruption setting are not exactly the same. For our proposed Multi-layer ETGS race method, the regret of each player $i, i\in[N]$ decomposes into two main components:
>
> 1. Exploration cost incurred by arm proposals in a round-robin fashion within each sub-phase.
>
> 2. Communication overhead between consecutive sub-phases, when the leader conveys the index of the selected ETGS layer to the other players through arm proposals.
>
> Both components of the regret depend on the choice of sub-phase length $d$. A larger $d$ increases the first component of the regret, while decreasing the second. By selecting the optimal value $d = \mathcal{O}(\sqrt{\log T})$, we can balance the trade-off betwen these two terms to minimize the total regret. Under this choice, the total regret and the communication cost can be regarded as having the same order of magnitude after omitting constant factors.
>
>
> >Q4：Upper and lower bound discussion.
>
> R: We thank the reviewer for pointing out this point. Yes, there is indeed a difference between the upper and lower bounds in how the preference gap is defined. Specifically, the upper bound is expressed in terms of the minimum preference gap $\Delta$ among the top-(N + 1) ranked arms for each player, while the lower bound is defined using $\tilde{\Delta}$, which in the Optimally Stable Bandit (OSB) instance corresponds to the gap between the stable arm and the next preferred arm. Since the player-optimal stable arm must be the first $N$ ranked, $\Delta$ is always smaller than $\tilde{\Delta}$, which also holds in the OSB setting. We have clarified this distinction and corrected our discussion of the upper and lower bounds in Remark 4.2 of the revised version.
>
> The mismatch between $\Delta$ and $\tilde{\Delta}$ is actually a fundamental open problem in the study of matching markets (even without corruption). It reflects the intrinsic difficulty and cost of achieving stable matchings through agent exploration and interaction. The best known lower bounds in the literature are established for the OSB setting and depend on the gap $\tilde{\Delta}$ (Sankararaman et al., 2021), but to the best of our knowledge, no existing algorithm has been proven to achieve an upper bound that depends directly on $\tilde{\Delta}$ in this sense.  Extending the analysis to improve the dependence on sub-optimality gap under corruption remains an interesting direction for future work.
>
> For the unknown $C$ setting, our upper bound contains an additional $\log^{2} T$ factor compared with the lower bound, and the two bounds also differ in their dependence on the sub-optimality gap. The multiplicative $\log^2 T$ factor arises because our method maintains $\log T$ ETGS instances to estimate the unknown $C$, and uses at most $\log T$ rounds for communication at the end of each sub-phase to achieve synchronization.  Determining whether these logarithmic factors can be removed—and thereby closing the gap to the lower bound—remains an important direction for future work.

---

> ### Author Response · Authors · 2025-11-20
> **Response to Reviewer wBB7**
>
> We sincerely appreciate the reviewer’s valuable comments. Below we provide point-by-point responses to each comment.
>
>
>
> >Q5: Explanation of each term in regret upper bound under unknown $C$ setting.
>
> R: In the original version of this submission, due to space constraints, we provided only the final form of the regret upper bound—that is, a bound in which terms of the same order were combined. In the revised version (the updated PDF), and in this response, we provide the full regret upper bound and explain the meaning of each term below.
> $$
>  Reg_i(T)\leq (N+N^2\log T+ 2NK\log T+N\log T)\Delta_{i,\max}+ 16K\log T \left(\frac{24\log T}{\Delta^2}+\frac{d\log T}{\Delta} \right)\cdot \Delta_{i,\max}$$
> $$ +16KC\left(\frac{24\log T}{\Delta^2}+\frac{d\log T}{\Delta} \right)\cdot \Delta_{i,\max}+ 16K\log T\left (\frac{24\log^2T}{d\Delta^2}+\frac{\log^2T}{\Delta}+\frac{24C\log T}{d\Delta^2}+\frac{C\log T}{\Delta}\right)\cdot \Delta_{i,\max}.
> $$
>
> 1. $(N+N^2\log T+2NK\log T+N\log T)\Delta_{i,\max}$. The term  $N\Delta_{i,\max}$  captures the regret in the first phase, when the algorithm is estimating each player’s identity/index. $N^2\log T \Delta_{i,\max}$ comes from the use of a decentralized Gale–Shapley (GS) algorithm across $\log T$ ETGS layers. $(2NK\log T+N\log T)\Delta_{i,\max}$  corresponds to the bad concentration events.
>
> 2. $16K\log T \left(\frac{24\log T}{\Delta^2}+\frac{d\log T}{\Delta} \right)\cdot \Delta_{i,\max}$. This is the regret contributed by those layers $\ell>\ell^*$ with sampling probability less than $1/C$.
>
> 3. $16KC\left(\frac{24\log T}{\Delta^2}+\frac{d\log T}{\Delta} \right)\cdot \Delta_{i,\max}$. This term reflects the regret mainly contributed  by those layers $\ell\leq \ell^*$ with sampling probability greater than or equal to $1/C$.
>
> 4. $ 16K\log T\left (\frac{24\log^2T}{d\Delta^2}+\frac{\log^2T}{\Delta}+\frac{24C\log T}{d\Delta^2}+\frac{C\log T}{\Delta}\right)\cdot \Delta_{i,\max}$. This term captures the regret contributed by the total communication cost incured across successive sub-phases.

---

### Official Review · Reviewer_dRww · 2025-10-31

**Soundness:** 2
**Presentation:** 2
**Contribution:** 3
**Rating:** 6
**Confidence:** 3

**Summary:**

This paper investigates an online learning problem in two-sidd decentralized matching markets where stochastic rewards are subject to malicious adversarial corruptions. The goal is for N players and K arms to identify the player optimal stable match whilst minimizing the pseudo-regret. The problem is first studied when the corruption level C is known, where the proposed algorithm achieves a regret upper bound that is shown to be near optimal with corresponding lower bounds. The authors extend to the setting of unknown C by coordinating exploration among $O(\log T)$ parallel running instances of their original algorithm. They validate their results with some empirical results as well.

**Strengths:**

1. The introduction of adversarial robustness into decentralized matching market bandits is a nice direction. The paper addresses realistic adversarial corruptions beyond the common stochastic feedback.

2. The multi-layer extension is elegant to solve the unknown C. The need to introduce a synchronization mechanism to overcome conflicts inherent in randomized strategies is a crucial observation.

3. The optimality analysis for the known C setting is a strong point.

4. The emprical validations are nice.

**Weaknesses:**

1. For the case of unknown C, there exists a suboptimality gap. The authors acknowledge this multiplicative logarithmic gap and attribute it to the overhead caused from running $\log T$ paralllel instances.

2. Some important details regarding the algorithmic procedures are deferred to the Appendix, specifically Appendix B; such as Phase 1 index estimation, Phase 3 decentralized GS, leader selection and synchronization specifics. I believe the papers novelty rests heavily on adapting decentralized mechanics to robustness. Therefore, even if the pseudo code remains supplemental, more insights should be provided in the main body of the paper.

**Questions:**

1. Is the multiplicative redundancy $\log^2 T$ on the regret bound of unknown C fundamentally unavoidable given the layer based approach? Is there a possibility of reducing the overhead to tighten the optimality gap, mye by using alternative communication strategies or nonuniform synchronization intervals?

2. In Remark 3.5, the optimal choice ofor hyper-parameter $d$ is theoretically derived as $O(\sqrt{\log T})$ to minimize the regret bound. Figure 1(c) shows that larger $d$ values worsen the empirical regret compared to $d=1$. Can you provide a deeper empirical discussion on the optimal $d$ value?
3. The upper bounds utilize the minimum preferene gap $\Delta$, while the lower bound utilizes $\tilde{\Delta}$ specific to Optimally Stable Bandits. Can you clarify the relationship between these two metrics? Does the near-optimality established in Remark 4.2 hold because $\Delta$ and $\tilde{\Delta}$ are equivalent under OSB? Is the derived lower bound also applicable to the general setting characterized by $\Delta$?

**Details Of Ethics Concerns:**

The authors provide an ethics statement indicating that the work is purely theoretical and does not involve human subjects, personal data, or foreseeable ethical concerns. I concur with this assessment.

---

> ### Author Response · Authors · 2025-11-20
> **Response to Reviewer dRww**
>
> We sincerely appreciate the reviewer’s valuable comments. Below we provide point-by-point responses to each comment.
>
> >Q1: Multiplicative $log^2⁡𝑇$ factor on the regret bound of unknown $C$ setting.
>
> R: We first clarify the origin of the $\log^{2} T$ factor in our upper bound for the unknown-corruption setting. One $\log T$ factor arises from maintaining $\log T$ ETGS layers to handle the unknown corruption budget $C$, analogous to the construction used in the single-player MLAER algorithm (Lykouris et al., 2018). The second $\log T$ factor comes from synchronization: because communication occurs only through arm proposals, informing all players of the leader’s chosen layer requires up to $\log T$ rounds. These two sources together result in the multiplicative $\log^{2} T$ factor.
>
> We also note that the communication component of the regret can indeed be reduced. By encoding both the time index and the arm index in the leader’s proposals, the target layer—chosen from ${1,2,\ldots,\log T}$ —can be communicated in at most $\log T / K$ rounds. Under this improved mechanism, the additional regret term decreases from $\log^{2} T$ to $\log^{2} T / K$.
>
> Whether the $\log^{2} T$ factor can be removed in decentralized matching markets remains open. Even in the single-player setting, eliminating the additional $\log T$ factor required substantial progress over several years. Early corrupted-bandit methods, such as MLAER [1] and BARBAR [2], inherently incurred a multiplicative$\log T$ term, and only more recent advances—most notably FTRL with Tsallis regularization [3]—managed to remove this dependence through carefully designed randomized and adaptive exploration strategies.
>
> However, extending these techniques to decentralized matching markets is far from straightforward. First, independent randomized sampling by multiple players inevitably leads to frequent conflicts, severely disrupting the matching dynamics and preventing the coordinated exploration necessary for learning stable matchings. Second, the FTRL-based approaches rely on adaptive exploration, whereas the state-of-the-art decentralized matching markets fundamentally depend on explicit and structured exploration procedures (e.g., ETC-style exploration) to ensure consistent interactions across players. These structural differences create substantial obstacles to directly transplanting single-player techniques into the multi-player matching setting.
>
> For these reasons, we believe that determining whether the $\log^{2} T$ factor can be completely eliminated in decentralized matching markets with unknown corruption is a genuinely challenging and unresolved question, and we regard this as an important and promising direction for future research.
>
> [1] Lykouris, T., Mirrokni, V. and Paes Leme, R., 2018, June. Stochastic bandits robust to adversarial corruptions. In Proceedings of the 50th Annual ACM SIGACT Symposium on Theory of Computing (pp. 114-122).
>
> [2] Gupta, A., Koren, T. and Talwar, K., 2019, June. Better algorithms for stochastic bandits with adversarial corruptions. In Conference on Learning Theory (pp. 1562-1578). PMLR.
>
> [3] Zimmert, J. and Seldin, Y., 2019, April. An optimal algorithm for stochastic and adversarial bandits. In The 22nd International Conference on Artificial Intelligence and Statistics (pp. 467-475). PMLR.
>
>
> >Q2: Writing and organization.
>
> R: Thank you for your helpful suggestions on the writing and organization of our paper. In the revised version (the updated PDF), we have added high-level explanations of these mechanisms to the main body of the paper.
>
>
> >Q3: Empirical discussion of the optimal hyper-parameter $d$.
>
> R: In the revision version we add new simulation experiments to explore the optimal choice of the hyper‑parameter $d$. Specifically, we evaluated our proposed multi‑layer ETGS race algorithm over $d \in \\{ 1, 2, 4, 6, 8 \\}$ for a total of $T = 500000$ rounds. These new experimental results appear in the updated PDF. Recall that our theory suggests the optimal sub-phase length should satisfy $d = \mathcal{O}(\sqrt{\log T}) \approx 4.3$ under $T=500000$. From the empirical results (Figure. 1(d) in the revision), we observe that settings with  $d = 4$ outperform the other choices, which aligns with our theoretical findings.

---

> ### Author Response · Authors · 2025-11-20
> **Response to Reviewer dRww**
>
> We sincerely appreciate the reviewer’s valuable comments. Below we provide point-by-point responses to each comment.
>
>
>
> >Q4：Different definition of suboptimality gap in lower bound.
>
> R: We thank the reviewer for pointing out this point. Yes, there is indeed a difference between the upper and lower bounds in how the preference gap is defined. Specifically, the upper bound is expressed in terms of the minimum preference gap $\Delta$ among the top-(N + 1) ranked arms for each player, while the lower bound is defined using $\tilde{\Delta}$, which in the Optimally Stable Bandit (OSB) instance corresponds to the gap between the stable arm and the next preferred arm. Since the player-optimal stable arm must be the first $N$ ranked, $\Delta$ is always smaller than $\tilde{\Delta}$, which also holds in the OSB setting. We have clarified this distinction and corrected our discussion of the upper and lower bounds in Remark 4.2 of the revised version.
>
> The mismatch between $\Delta$ and $\tilde{\Delta}$ is actually a fundamental open problem in the study of matching markets (even without corruption). It reflects the intrinsic difficulty and cost of achieving stable matchings through agent exploration and interaction. The best known lower bounds in the literature are established for the OSB setting and depend on the gap $\tilde{\Delta}$ (Sankararaman et al., 2021), but to the best of our knowledge, no existing algorithm has been proven to achieve an upper bound that depends directly on $\tilde{\Delta}$ in this sense.  Extending the analysis to improve the dependence on sub-optimality gap under corruption remains an interesting direction for future work.

---

### Official Review · Reviewer_UUKf · 2025-11-02

**Soundness:** 3
**Presentation:** 3
**Contribution:** 3
**Rating:** 6
**Confidence:** 2

**Summary:**

The paper proposes a multilayer preference learning bandit algorithm that can adapt to an unknown level of corruption. Numerical experiment corroborated the theoretical findings and shows an advantage over existing approaches.

**Strengths:**

I find the paper to be well written, with a clear storyline, strong motivation, and sufficient explanation of the technical challenges. In particular, the authors effectively use concrete examples to illustrate why a two-sided market with corrupted feedback is an interesting and relevant setting. The discussion of the technical difficulties and the insights provided is also sufficiently detailed and informative.

**Weaknesses:**

- From the paper itself, it is not entirely clear how the algorithm's overall computation complexity scales as $K$ and $T$ grow. In particular, it is unclear how well the monitoring subroutine scales compared to existing approaches for online learning in a two-sided market.
- The authors commented on adversarial feedback. Would an FTRL/FTPL-type algorithm achieve a best-of-three-worlds guarantee in the sense that it achieves the same regret in all stochastic, corrupted, and adversarial environments? Can the authors comment on the related works in this direction and outline the technical difficulties in unknown corruption level, and why this is not a suitable algorithmic framework to consider for this setting?

**Questions:**

See the previous section

---

> ### Author Response · Authors · 2025-11-20
> **Response to Reviewer UUKf**
>
> We sincerely appreciate the reviewer’s valuable comments. Below we provide point-by-point responses to each comment.
>
> >Q1: Computation complexity.
>
> R: We discuss the overall computational complexity of the multi-layer ETGS race method below.
>
> 1. During each sub-phase of the exploration phase, each player simply follows the round-robin pattern associated with the current layer $\ell$ to propose arms and update the statistics $\hat{\mu}^{\ell}_ {i,j}$ and $T^{\ell}_ {i,j}$. The computational cost for proposing an arm and updating the corresponding statistics is $\mathcal{O}(1)$.
>
> 2. We now review the workflow of the monitoring subroutine. At the end of each sub-phase, to decide whether exploration at the current layer is complete, each player checks whether all other players have already estimated their preference ranks. This may require scanning over the top arms, which in the worst case incurs a cost of $\mathcal{O}(K)$, since the player may need to inspect up to $K$ arms.
>
> Hence, all operations, including arm proposals and statistics updates (each with cost $\mathcal{O}(1)$), together with the monitoring step of cost $\mathcal{O}(K)$, impose only a modest computational burden on each player. Consequently, the overall computational complexity of the multi-layer ETGS race method remains low. This is because although the method runs $\log T$ parallel ETGS instances to handle the unknown corruption level, in each sub-phase players only need to execute operations within a single corresponding ETGS instance.
>
>
> >Q2: FTRL/FTPL-type algorithm.
>
> R: First, we appreciate the reviewer's suggestion to include studies on FTRL/FTPL-type algorithms that obtain best-of-three-worlds guarantees. In the revised version (updated PDF), we have added the relevant works in the related-work section of the appendix.
> FTRL/FTPL-type algorithms does have achieved best-of-three-worlds guarantees in several single-player online learning settings such as multi-armed bandits, linear bandits, and graph-feedback bandits. However, extending these techniques to our multi-player decentralized matching-market setting faces fundamental obstacles, as we outline below.
>
> 1. Algorithmic level. In single-player bandit problems, FTRL/FTPL algorithms rely on injecting randomness into action selection, preventing an adversary from exploiting deterministic behavior. In contrast, in a decentralized matching market each player independently samples an arm to propose using its own FTRL/FTPL-type policy. Such independent randomization creates a challenge absent in single-player settings: Multiple players may propose the same arm in a round, resulting in “collisions” that severely degrade exploration efficiency. The randomness that protects a single learner from adversarial feedback now induces harmful interference across players. These collisions disrupt the learning dynamics of the entire system, and therefore a straightforward application of FTRL/FTPL is not suitable for decentralized matching markets.
>
> 2. Theoretical level. The BoTW guarantee requires a gap-independent regret analysis. However, to the best of our knowledge, even in the pure stochastic matching-market setting, there lacks an adaptive bandit algorithm with gap-independent regret bounds. The technical difficulty stems from the structural properties of matching: small perturbations in reward means can lead to dramatically different stable matchings due to multi-player interaction. This discontinuity stands in sharp contrast to single-player bandits, where regret can typically be related continuously to estimation error. As a result, the analytical tools used for FTRL/FTPL in single-player settings do not extend to matching markets. Achieving a BoTW guarantee via FTRL would first require overcoming this unresolved gap-independent challenge, which is itself a major open problem.

---

### Official Review · Reviewer_qjPx · 2025-11-03

**Soundness:** 3
**Presentation:** 3
**Contribution:** 2
**Rating:** 6
**Confidence:** 3

**Summary:**

This tackles the problems of one-to-one decentralised matching with bandit feedback  under adversarial corruption, for the case where one side's (arms) preferences are known.

The decentralised matching works by players proposing to arms, and each arm accepting their most prefered player. The feedback here is corrupted, with stochastic rewards $r^S$ being modified to $r$ up to a total corruption budget $C$ over the horizon $T$.

The basic algorithmic idea is to take the corruption level into account when creating confidence intervals. This is first analysed for known corruption in 3.1, which results in $KC/\Delta$ additional regret and in 3.2 for the case of unknown corruption.

The paper refers to an index estimation phase in some algorithms, to take care of conflicts. I personally find the decentralised setting to just add unnecessary complexity, without making the problem more interesting. If you have to force everybody to do a round-robin according to some indices, why not just force them to play according to some centrally defiend schedule? This would also avoid the complications with sampling in the unknown corruption case.

**Strengths:**

+ It is a natural extension of the stochastic case.
+ The proof is not a completely straightforward adaptation of Lykouris to matching.

**Weaknesses:**

- Maybe the decentralised setting is too complicated.

**Questions:**

? The paper refers to an index estimation phase in some algorithms, to take care of conflicts. I personally find the decentralised setting to just add unnecessary complexity, without making the problem more interesting. If you have to force everybody to do a round-robin according to some indices, why not just force them to play according to some centrally defiend schedule? This would also avoid the complications with sampling in the unknown corruption case. But maybe then the proofs would be too straightforward from Lykouris.

---

> ### Author Response · Authors · 2025-11-20
> **Response to Reviewer qjPx**
>
> We sincerely appreciate the reviewer’s valuable comments. Below we provide the corresponding responses.
>
> >Q: Motivation for the decentralized matching market setting.
>
> R:  First, we note that if we require each player to play according to a centrally defined schedule, a central platform is needed to coordinate all players throughout the learning process. Below, we clarify the challenges of deploying such centralized matching markets in realistic applications, and we motivate the study of decentralized matching-market bandit learning by explaining why this formulation is both natural and practically relevant.
>
>
> 1. Decentralization as the Norm in Real-World Matching Platforms. The decentralized learning framework—where no central node aggregates all information—is widely used in multi-agent settings such as distributed optimization and multi-agent bandits, and better reflects practical deployment constraints. In matching markets, decentralization is not just a theoretical alternative but the prevailing mode of operation. Many modern platforms, including online labor and crowdsourcing markets, lack a central clearinghouse and do not support global information sharing. Agents typically observe only limited, local feedback, making the decentralized formulation a natural fit for these environments [1].
>
> 2. Privacy and Transparency Considerations. Privacy and transparency concerns often rule out the use of a central coordinator. Sharing observation with a central arbiter is prone to privacy breach, lacks transparency of the arbiter, and is susceptible to untruthful inputs from agents. In contrast, decentralization better respects privacy: each agent learns and acts independently, without disclosing sensitive information [2].
>
> 3. For scalability reasons as well, decentralized solutions are often preferred. In specific, decentralized learning allows agents to compute locally, making algorithms scalable to large, dynamic populations [3].
>
> Therefore, we believe that studying bandit learning in decentralized matching markets with adversarial corruption is both more practical for real-world deployment and more significant for advancing the theoretical understanding of multi-agent bandit learning.
>
> [1] Liu, L.T., Ruan, F., Mania, H. and Jordan, M.I., 2021. Bandit learning in decentralized matching markets. Journal of Machine Learning Research, 22(211), pp.1-34.
>
> [2] Rees-Jones, A. and Skowronek, S., 2018. An experimental investigation of preference misrepresentation in the residency match. Proceedings of the National Academy of Sciences, 115(45), pp.11471-11476.
>
> [3]  Larsson, S., 2018. Law, society and digital platforms: Normative aspects of large-scale data-driven tech companies. In The RCSL-SDJ Lisbon Meeting 2018" Law and Citizenship Beyond The States”.

---

### Author Response · Authors · 2025-12-01
**Official Comment by Authors**

Dear Area Chair,

We are sincerely grateful to you and all the reviewers for the time and effort dedicated to reviewing our paper. We deeply appreciate all reviewers for their thoughtful and constructive comments, which have significantly improved both the experimental depth and theoretical clarity of the paper. Notably, all four reviewers expressed an overall **positive assessment** of the submission. Their main requests focused on additional **discussion and clarification on certain definitions and technical details.**

In response, we have provided point-by-point replies to every comment and thoroughly revised the paper. We believe that these responses and revisions better highlight our main contributions and comprehensively address the reviewers’ concerns. Below we provide an overview of our responses to the reviewers’ main concerns.



## Summary of Author Responses

1. **Motivation for decentralized matching markets (Reviewer qjPx).** We explain that the decentralized formulation is a natural choice in practical deployments, and we clarify how it benefits privacy and scalability in realistic matching market applications.

2. **Discussion on FTRL/FTPL-type algorithms (Reviewer UUKf).**  We explain why extending FTRL/FTPL-type methods to decentralized matching markets with adversarial corruption encounters fundamental obstacles, in both algorithmic and theoretical aspects. In the revised manuscript, we also add a dedicated discussion of FTRL/FTPL-type algorithms in the related work section.

3. **Multiplicative logarithmic factor on the regret bound in the unknown $C$ setting (Reviewer dRww).** We first clarify the source of this logarithmic factor. Then, we explain how an improved communication strategy reduces the factor from $\log^2 T$ to $\log^2 T / K$. Finally, we provide a detailed explanation of why removing this factor remains an open problem in decentralized matching market settings.

4. **Empirical discussion of the optimal hyper-parameter $d$ (Reviewer dRww).**  We add new simulation experiments in the revision, evaluating our proposed multi-layer ETGS race algorithm over a more finely grained range of the hyper-parameter $d$. The empirical results are consistent with our theoretical findings, indicating that the optimal $d$ scales as $\mathcal{O}(\sqrt{\log T})$.

5. **Clarification on preference gaps in the upper and lower bounds (Reviewers dRww, wBB7).** Regarding the preference gaps $\Delta$ and $\tilde{\Delta}$ used in the upper and lower bounds, we first clarify their respective definitions and their relationship. In the revised version, we clearly distinguish these gaps and correct the discussion about the upper and lower bounds in Remark 4.2. We also explain that the mismatch between  $\Delta$ and $\tilde{\Delta}$ is in fact a fundamental open problem in the study of matching markets, even in the absence of corruption.



Once again, we sincerely thank the reviewers for their insightful and positive feedback, and thank you for the time and dedication in the review process.



Best regards,

The Authors of Submission 4211

---

### Meta-Review · Area_Chair_UJVi · 2025-12-08

**Summary:**

Even in the initial reviews, the reviews had a clear consensus for acceptance.  The minor concerns are adequately addressed in the response, so I can only assume the scores would have remained positive, or potentially even increased slightly. The reviewers commented on how combining robustness with markets is interesting and non-trivial, and commented positively on both the theory and experiments.

**Reviewer Concerns:**

As noted above, there are no obvious remaining concerns significant enough to impact the decision.

**Reviewer Scores:**

6/6/6/6 is already a very good set of scores, and it's possible that one or more would raise to 8.

---

### Decision · Program_Chairs · 2026-01-26

Accept (Poster)